

**OH measurements in the coastal atmosphere of South China: missing OH sinks in aged air**
**masses**
Zhouxing Zou[1], Qianjie Chen[1], Men Xia[1], Qi Yuan[1], Yi Chen[2], Yanan Wang[1], Enyu Xiong[1], Zhe Wang[2],
and Tao Wang[1*]
[1]Department of Civil and Environmental Engineering, The Hong Kong Polytechnic University, Hong
Kong, China
[2]Division of Environment and Sustainability, The Hong Kong University of Science and Technology,
Hong Kong, China
Correspondence to:
Tao Wang (tao.wang@polyu.edu.hk)
**Abstract**
The hydroxyl radical (OH) is the main oxidant responsible for the removal of many reduced trace gases
and the formation of secondary air pollutants. However, due to technical difficulties in measuring OH,
the existing measurements of atmospheric OH concentrations are limited, and its sources and sinks are
not well understood under low $NO_x$ conditions. In this study, we observed the OH concentrations using
chemical ionization mass spectrometry at a coastal site in Hong Kong from October to November 2020.
The average noontime OH concentration over the study period was measured at $4.9 \pm 2.1 \times 10^6$ $cm^{-3}$.
We found that a box model with comprehensive observational constraints reproduced the observed
daytime OH concentrations when air parcels originated from the continental regions. However, this
model overpredicted the observed daytime OH concentrations for coastal air parcels by 73% on average.
Missing OH reactivity is proposed to be the cause of this overprediction. High missing OH reactivity
was found in the case of low concentrations of nitrogen oxides ($NO_x$) and volatile organic compounds,
as well as in aged air, suggesting that there could be unmeasured chemical species that cause the model
to overestimate OH in aged coastal air parcels. Further studies are needed to identify these unmeasured
chemical species and their contributions to the OH budget, in order to better quantify the formation of



secondary air pollutants.
**1.  Introduction**
The hydroxyl radical (OH) dominate atmospheric oxidative capacity and participates in nearly all sunlit
tropospheric chemistry. The primary sources of the ambient OH radical include the photolysis of ozone
($O_3$) and nitrous acid (HONO) and the ozonolysis of alkenes. OH sinks are mainly the reactions of OH
with trace gases, including carbon monoxide (CO), sulfur dioxide ($SO_2$), nitric oxide (NO), nitrogen
dioxide ($NO_2$), methane, and other volatile organic compounds (VOCs; Fuchs et al., 2018).
Heterogeneous uptake of OH represents very minor OH sinks (Ivanov et al., 1996). In reactions with
CO and VOCs, peroxy radicals ($HO_2$ and $RO_2$) are produced and then recycled back into OH in the
presence of NO as a secondary OH source. This interconversion is closely related to photochemical
smog production (Stone et al., 2012). The reaction of OH with $SO_2$ and $NO_2$ produces $H_2SO_4$ and $HNO_3$,
contributing to new particle formation and the acidity of rain, fog, and aerosols. OH also plays an
important role in the climate system through reactions with the greenhouse gas $CH_4$ and the sulfate
aerosol precursor dimethyl sulfide (DMS)(Berresheim, 2002).
Measuring ambient OH is challenging due to its high reactivity, short lifetime (< 1 s), and low
environmental concentration (Stone et al., 2012). After decades of efforts, tropospheric OH radicals can
now be detected following the development of laser-induced fluorescence (LIF)–fluorescence assay
with gas expansion (Hard et al., 1984), chemical ionization mass spectrometry (CIMS; Eisele et al.,
1991), and open-path differential optical absorption spectrometry (Hausmann et al., 1997). The theory,
advantages, and disadvantages of various measuring techniques have been discussed previously (Hard
et al., 1979; Mao et al., 2012). Using these techniques, multiple campaigns have been conducted to
measure the atmospheric OH concentrations in different regions around the globe. Figure 1 summarises
the previous field observations of OH radicals in various environments.
OH observations are often compared with model simulations to evaluate whether a model has included
the major OH sources and sinks. A summary of the results of the most recent studies is shown in Table
1 with the simulation to observation ratios ($R_{S/O}$). As concluded in previous reviews (Stone et al., 2012;



Rohrer et al., 2014; Lu et al., 2019a), observed OH concentrations can generally be reproduced by box
models under high NO conditions (NO > 1 ppb), such as at urban sites or within polluted air masses
(Shirley et al., 2006; Griffith et al., 2016; Slater et al., 2020). However, discrepancies between model
predictions and observations have often been found under low NO conditions (NO < 1 ppb). The model
typically overpredicts OH concentrations in a low VOC environment and underpredicts them in a high
biogenic VOC (BVOC) environment, as discussed below.
The model overestimation of OH has been found in various environments, including remote marine
boundary layers and coastal, urban, and Arctic regions (Table 1). Previous studies have attributed the
discrepancy to the overestimation of OH sources, missing OH sinks, and the uncertainties inherent in
model simulation and observation. For example, model overestimation of OH has been found when
dominant sources, such as HCHO and NO (Zhang et al., 2006), HONO (Kukui et al., 2014), and $HO_2$
(Kanaya et al., 2007a), are overestimated. In these cases, the overestimation of OH was resolved when
these sources were better constrained in the model. Unmeasured VOCs have been proposed as the
missing OH sinks, especially in aged air (McKeen et al., 1997; Carslaw et al., 1999; Berresheim, 2002;
Creasey et al., 2003; Mauldin et al., 2010; Griffith et al., 2016). Previous studies have shown evidence
of missing OH sinks in the forest (Hansen et al., 2014) and marine (Thames et al., 2020) regions, likely
resulting from unmeasured organic compounds in biogenic ((Kaiser et al., 2016)) or oceanic (Thames
et al., 2020) emissions and their oxidation products. In relation to potential overestimations caused by
simulation and measurement uncertainties, some studies have shown that the overestimations fall within
measurement uncertainties (McKeen et al., 1997, Carslaw et al., 1999), while others have suggested a
possible sampling loss of OH (Mauldin et al., 2010) or a possible calibration bias due to low relative
humidity (Mauldin et al., 2001).
Underestimations of OH by models have mostly been found in forest areas with high BVOC emissions
(mostly isoprene) and low NO conditions. These underestimations have usually been attributed to
missing OH sources (Tan et al., 2001; Lelieveld et al., 2008; Hofzumahaus et al., 2009; Whalley et al.,
2011). To explain the missing sources, a series of new OH regeneration reactions under low NO
conditions were proposed based on chamber experiments that investigated the oxidation of isoprene by



OH. This mechanism, known as the Leuven isoprene mechanism (LIM1; Peeters et al., 2014), includes
unimolecular reactions (Peeters et al., 2009; Da Silva, 2010; Fuchs et al., 2013; Novelli et al., 2020)
and isomerisation of isoprene and/or its products (Peeters and Müller, 2010; Fuchs et al., 2014). With
the adoption of this mechanism, the simulated OH concentration increased by 20%–30% in the forest
region (Lew et al., 2020). Another breakthrough was the development of a new chemical scavenging
technique in LIF instruments that was able to determine the interference to the instrument's background.
Some studies have shown that the interference in LIF instruments can partly explain the previously
observed high OH concentrations (Mao et al., 2012; Hens et al., 2014; Novelli et al., 2014a; Feiner et
al., 2016; Woodward-Massey et al., 2020). With the adoption of interference scavenging and the LIM1
improved mechanism, measurements using LIF in an Alabama forest (Feiner et al., 2016) and CIMS in
Amazon forests (Jeong et al., 2022) agreed with the OH concentration predicted by the model. However,
the models in other studies continued to underestimate OH with the improved mechanism (Tan et al.,
2019; Lew et al., 2020).
The industrialisation of the Pearl River Delta (PRD) region of south China over the past three decades
has been accompanied by high anthropogenic emissions of air pollutants (Lu et al., 2013b), causing
elevated concentrations of surface ozone (Wang et al., 2019b) and particulate matter (Yao et al., 2014).
Measurements of OH in the PRD region that were taken using LIF at a forested site (Backgarden)
indicated missing OH sources at this BVOC-rich site (Hofzumahaus et al., 2009; Lu et al., 2012). More
recently, OH concentrations were measured at a suburban site in Shenzhen during autumn 2018 (Wang
et al., 2019a; Wang et al., 2021) using a newly developed LIF instrument. The OH concentrations, which
had an average value of $5.3 \times 10^6$ cm$^{-3}$ around noon, were briefly presented with no comparison to
modeled OH.
In the present study, we measured OH concentrations using quadrupole CIMS from October to
November 2020 at a background site in Hong Kong. The study aimed to determine the OH
concentrations in coastal south China and to investigate whether they could be simulated by a state-of-
the-art chemical model under different airflow conditions. We first give a brief description of the site
and OH measurement procedure, including the working theory of CIMS, calibration, uncertainties, and



modelling setup. We then present the overall measurement results for different air masses and compare
them with those found in previous studies. After this, we simulate OH concentrations using a box model
constrained by comprehensive observations and discuss possible reasons for the model–observation
discrepancy. Our measurements add to the limited database of ambient OH radical concentrations, while
our analysis sheds light on possible missing OH sinks under low $NO_x$ conditions.
**2. Methodology**
**2.1 The Hok Tsui Supersite**
Our field campaign was conducted at the Cape D'Aguilar (also known as Hok Tsui, HT) Air Quality
Supersite, which is operated by the Hong Kong Environmental Protection Department, between 6
October and 24 November 2020. The HT Supersite (22°12'32" N, 114°15'12" E) is a coastal site located
at the south-eastern tip of Hong Kong Island. The site is surrounded by ocean, vegetation, and scattered
country roads (Figure 2) and is around 15 km away from the nearest urban centre. There is no strong
anthropogenic emission source in the surrounding area apart from the ocean-going vessels travelling in
nearby waters (Peng et al., 2022). Nonetheless, the site does occasionally receive polluted air masses
from mainland China, including masses from the highly urbanised PRD region (Li et al., 2018; Peng et
al., 2022).
We measured OH radicals, $O_3$, $NO_x$, CO, HONO, VOCs, oxygenated VOCs (OVOCs), relative humidity,
temperature, $NO_2$ photolysis frequency ($J_{NO2}$), and aerosol size distribution. Table 2 summarises the
measurement technique, resolution, and detection limits. The OH-CIMS was housed in an air-
conditioned shelter in yard B together with the time-of-flight (ToF)-CIMS and ozone and $NO_x$ analysers
(Figure 2). $J_{NO2}$ was measured on top of the shelter. The other species and the aerosol size distribution
were measured inside the main station building in yard A, which was located around 10 m away from
yard B (Figure 2). The backward trajectory was calculated at 1-hour intervals on sampling days at an
elevation of 60 m above ground level using the MeteoInfoMap software package (meteothink.org, Wang,
2014; Wang, 2019).



**2.2 OH radical measurements**
OH radical concentrations were indirectly measured using a custom-built quadrupole CIMS instrument.
CIMS was originally developed by Eisele and Tanner (1991) and improved upon in subsequent works
(Eisele and Tanner, 1993; Tanner and Eisele, 1995; Tanner et al., 1997). Detailed descriptions of CIMS
measurement principles, configuration, optimisation, and calibration have been provided by Pu et al.
(2020); here we only give a summary. The details of the technique, such as flow rates, reaction time,
and concentrations, are shown in Table 3.
A schematic diagram of our OH-CIMS instrument is shown in Figure 3. The procedure was as follows.
First, ambient air was drawn into the stainless steel inlet with a turbulence-reducing scoop by the inlet
pump. The central part of the air in the stainless steel inlet was then drawn into the sample inlet, where
OH was converted into $H_2SO_4$ by adding $SO_2$ to the sample flow. In the sheath flow, the $NO_3^-$ reagent
ion cluster was produced by passing an $HNO_3^-$-containing flow through a $^{210}Po$ ion source. The
converted $H_2SO_4$ in the sample flow was then reacted with the excess $NO_3^-$ cluster in the sheath flow
and converted into an $HSO_4^-$ ion cluster in the ionization chamber. The $NO_3^-$ and $HSO_4^-$ ion clusters
further dissociated in the collisional dissociation chamber (CDC), refocused in the ion guide chamber
(IGC), and were then detected by the detector in the ion detection chamber (IDC). In this case, $HSO_4^-$
and $NO_3^-$ were detected by the peak intensities at m/z = 97 ($S_{97}$) and m/z = 62 ($S_{62}$). The $HSO_4^-$ ion
concentration was determined based on relative signal strength (the $S_{97}/S_{62}$ ratio) rather than absolute
signal ($S_{97}$; Berresheim et al., 2000).
It should be noted that interference gases, such as ambient $H_2SO_4$, Criegee intermediates (Berresheim,
2002; Mauldin et al., 2012; Novelli et al., 2014b), and artificial OH produced by the ion source, also be
converted into $HSO_4^-$ and contribute to the signal $S_{97}$. To mitigate such interference, the scavenger gas
($C_3F_6$) and $N_2$ were added to the sample flow through electrically operated valves (see the pulsed flow
in Figure 3) that automatically switched injection positions every 1 min. When a scavenger gas is added
to the front injectors, ambient OH radicals are eliminated by the scavenger instead of reacting with $SO_2$
due to the higher concentration (~100 times) and faster reaction (Dubey et al., 1996) of $C_3F_6$ than of
$SO_2$ in the sample flow. This allows the background signal ($BS_{97}$) contributed by the interference gases



to be determined. When a scavenger gas is added to the rear injector, the ambient OH radicals and
interference gases react with $SO_2$ to give the total signal ($TS_{97}$). Then, the ambient OH signal can be
obtained by subtracting the signal contributed by interference ($BS_{97}$) from the total signal ($TS_{97}$). The
measured OH concentration ([OH]) can be calculated using the following equation (E1):

$$[OH] = \frac{1}{c} \times \frac{TS_{97} - BS_{97}}{S_{64}} \text{ (E1)}$$

where $c$ is the calibration factor obtained from calibration that was performed using the calibrator shown
in Figure 3. The calibration is based on the production of OH radicals through the photolysis of water
vapor by 184.9 nm light in the airflow through the calibrator (E2).

$$H_2O + h\nu(184.9\ nm) \rightarrow H + OH \text{ (E2)}$$

$$H + O_2 + M \rightarrow M + HO_2 \text{ (E3)}$$

$$[OH] = [H_2O] * \sigma_{H_2O} * \Phi * It \text{ (E4)}$$

The OH concentration produced by the calibrator is calculated by E4. $\sigma_{H_2O}$ (= $7.22 \times 10^{-20}$ cm$^2$; Cantrell
et al., 1997) is the photolysis cross-section of water vapor, while $\Phi$ represents the photolysis quantum
yield, which is assumed to be 1 (Kürten et al., 2012). The $H_2O$ concentration was calculated using the
measured temperature and dew point temperatures of the calibrating air using the ideal gas law. The
photon flux ($It$) was determined using the chemical actinometry method (Kürten et al., 2012). In this
study, the $It$ values were measured before and after the field campaign and no significant difference
was found. Calibration was carried out at least every two days during the campaign, as well as before
and after any changes in settings. The difference in calibration factors was included by the calibration
accuracy.
The overall calibration accuracy was estimated at 38%, by calculation that took into account the
uncertainty of all of the parameters measured in E4 during the calibration process and the variation in
calibration factors during the campaign. The detection limit is approximately $1.5 \times 10^5$ cm$^{-3}$ (signal-to-
noise ratio of 2) in the laboratory. However, due to variation in the concentrations of $H_2SO_4$ and other
interference gases in the ambient air, the detection limit for *in situ* measurement may change along with
the ambient conditions and higher than the detection limit in the laboratory. The daytime and nighttime
average detection limits in this campaign were 1.0 and $0.7 \times 10^6$ cm$^{-3}$, respectively.



**2.3 Box modelling**

The Framework for 0-D Atmospheric Modelling (F0AM) using the Master Chemical Mechanism (MCM) v3.3.1 (Wolfe et al., 2016) was used to simulate OH concentrations. MCM v3.3.1 (http://mcm.leeds.ac.uk/MCM) is a near-explicit chemical mechanism that includes over 17,000 elementary reactions of 6700 primary, secondary, and radical species (Jenkin et al., 2015). The isoprene degradation mechanisms, and in particular the $HO_x$ recycling mechanisms, are included in MCM v3.3.1. The MCM mechanism has been used in previous studies to investigate OH chemistry in different environments, including forests (Stone et al., 2011), urban areas (Slater et al., 2020), suburban areas (Tan et al., 2018), and coastal regions (Sommariva et al., 2004). Observational data (shown in Table S1) were used to constrain the model. These data included VOCs, OVOCs, $SO_2$, $NO_x$, CO, $O_3$, HONO, and meteorological parameters (temperature, relative humidity, pressure, and $J_{NO2}$). The photolysis frequencies for other species were calculated as a function of the solar zenith angle ($\varphi$) using E5 and then scaled using observed versus modelled $J_{NO2}$.

$$J = l \times (cos\varphi)^m \times e^{-n \times sec\,\varphi} \text{ (E5)}$$

The values of the $l$, $m$, and $n$ parameters for photolysis frequency with respect to different species were drawn from Saunders et al. (2003).

In this study, the first-order physical loss process, with a 24-hour lifetime for all species, was included in the model to represent physical processes such as advection, deposition, and dilution (Wolfe et al., 2016; (Chen et al., 2022)). The physical loss process has a negligible influence on OH simulation because the OH concentrations are controlled by fast *in situ* chemistry.

The heterogeneous uptake of $HO_2$ by aerosols was included in the model by assuming a pseudo-first-order loss of $HO_2$ (E6–E8; (Jacob, 2000):

$$\frac{d[HO_2]}{dt} = -k_{HO_2}[HO_2] \text{ (E6)}$$

$$k_{HO_2} = \frac{V_{HO_2} \times S_a \times \gamma_{HO_2}}{4} \text{ (E7)}$$

$$V_{HO_2} = \sqrt{\frac{8RT}{\pi \times MW_{HO_2}}} \text{ (E8)}$$





where $k_{HO_2}$ is the first-order loss rate coefficient of HO$_2$ by aerosol uptake, $\gamma_{HO2}$ is the effective HO$_2$
uptake coefficient (= 0.1 for the base model run; (Guo et al., 2019), $V_{HO_2}$ is the mean molecular velocity
of HO$_2$, $S_a$ is the aerosol surface area concentration measured by scanning mobility particle sizing
(SMPS), and $MW_{HO_2}$ (= 17 g/mol) is the molecular weight of HO$_2$. We assumed in the model that the
products of heterogeneous HO$_2$ loss would not participate in further reactions (Guo et al., 2019).
The observation data were averaged every 10 mins for the model input. Any missing values were
calculated assuming linear interpolation. The measured concentrations of NO and NO$_2$ were used to
constrain the model, with the NO/NO$_2$ ratio calculated based on the family conversion, as recommended
in a previous study (Wolfe et al., 2016; Figure S1). Due to the clean condition of the coastal air, some
of the reactive alkenes and long-chain alkanes were below detection limits. For the simulation of those
compounds, we used concentrations that were half of the detection limits. The measured VOCs were
further divided into those of anthropogenic origin (AVOCs) and biogenic origin (BVOCs). The AVOCs
included alkanes (C$_2$–C$_8$), benzene, and TEXs (toluene, ethylbenzene, and xylenes), which covered the
dominant species originating from petroleum gas and industrial solvent evaporation (Tang et al., 2008),
while the BVOCs included isoprene, terpene, pinene, and limonene. The majority (> 95%) of the
measured OVOCs in this study were C$_1$–C$_3$ aldehydes, ketones, and acids. For each run, a three-day
spin-up was performed to create a stable model environment and to avoid the uncertainty of
unconstrained species (Carslaw et al., 1999).
**3.   Results and Discussion**
**3.1 Overview of observations**
Figure 4 shows the time series of observed OH concentrations, along with the concentrations of other
trace gases and the meteorological parameters, during the campaign. The weather conditions during the
study featured relatively high temperatures, high relative humidity (RH), and strong solar radiation,
consistent with previously reported autumn observations at the same site (Li et al., 2018; Peng et al.,
2022). The air temperature ranged from 20°C to 30°C and RH ranged from 40% to 96%. The photolysis
frequency of NO$_2$ ($J_{NO2}$) peaked at $8 \times 10^{-3}$ s$^{-1}$ around noon on sunny days but decreased to $2 \times 10^{-3}$ s$^{-1}$
on cloudy days. The observed OH concentrations were mostly above the detection limit during the

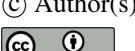



daytime but fell closer to the detection limit at night. The OH concentrations showed a distinct diurnal
pattern and a positive correlation with $J_{NO2}$ ($R^2$ = 0.67, Figure S2). The daily maximum OH
concentration varied from $2.1 \times 10^6$ $cm^{-3}$ on 21 November, accompanying the lowest level of solar
radiation, to $15.4 \times 10^6$ $cm^{-3}$ on 7 November which was measured during a pollution episode. The
pollution episode began on the evening of 6 November and featured a maximum concentration of 174.0
ppb $O_3$, 8.7 ppb NO, 22.7 ppb $NO_2$, 36.5 ppb total VOCs, and 48.1 ppb total OVOCs. The OH
concentration peaked the next day (7 Nov). This suggests abundant OH sources and fast radical
propagation under high-$NO_x$ and high-VOC conditions.
Figure 5 shows the average diurnal profiles of OH and other representative species. On average, the
maximum OH concentration was $4.9 \pm 2.1 \times 10^6$ ($1\sigma$) $cm^{-3}$. As shown in Table 1, the OH concentrations
at our site were comparable to those reported in previous field studies conducted at tropical coastal sites.
For example, the reported OH maximum concentration was $4.5 \times 10^6$ $cm^{-3}$ in the low-altitude remote
tropical troposphere (Brune et al., 2020). In a study conducted in autumn at a suburban site in Shenzhen,
approximately 50 km away from our site, an OH diurnal maximum concentration of $5.3 \times 10^6$ $cm^{-3}$ was
observed (Wang et al., 2021). Figure 5 also shows the average diurnal patterns of the other trace gases
measured. The primary precursor of OH, HONO, peaked in the morning at $0.21 \pm 0.09$ ppb around 7:00
local time (LT), while $O_3$ peaked in the afternoon ($70 \pm 20$ ppb at around 16:00 LT). The average NO
and $NO_2$ concentrations reached a maximum of $1.2 \pm 1.6$ ppb at around 10:00 LT and $4.9 \pm 3.2$ ppb at
around 18:00 LT, respectively. Isoprene showed a diurnal pattern similar to that of $J_{NO2}$ and OH, peaking
at $0.5 \pm 0.4$ ppb at noon. The average concentrations of all of the measured species during the campaign
are shown in Table S1.
Figure 6 shows the hourly backward trajectories over the whole campaign. Consistent with previous
studies conducted at HT in the same season (Li et al., 2018; Peng et al., 2022), the air masses were
dominated by continental air masses containing high concentrations of pollutants (Figure 6a) and less
polluted coastal air masses (Figure 6c). In this campaign, we did not encounter oceanic air masses from
the south. The average noontime OH concentration was $5.0 \pm 2.2 \times 10^6$ $cm^{-3}$ in the continental air mass
cluster (Figure 6b) and $3.3 \pm 1.6 \times 10^6$ $cm^{-3}$ in the coastal air (Figure 6d).



**3.2 Model–observation comparison**

To investigate the performance of the MCM box model in simulating OH chemistry at our site, we selected 4 days featuring the continental air mass (8, 21, 22, and 23 Oct) and 4 days featuring the coastal air mass (25–27 Oct, 5 November). We also selected 10 October as a specific case due to the shifting continental and coastal air masses within the same day during the daytime. These days were selected for model analysis because they comprised relatively complete chemical data that could be used to constrain the model.

**3.2.1 Selected continental air mass cases**

Figure 7 shows the comparison between the simulated and observed OH concentrations for the selected cases in the continental and coastal air masses (4 days each). The simulated OH concentrations of the four continental cases (8 October and 21–23 October) were mostly within the OH measurement uncertainty ($2\sigma$), with a daytime average $R_{S/O}$ of 0.97 (Figure 8) and a range from 0.87 to 1.02 (Figure 7). High $NO_x$ (~ 5 ppb) and VOCs (~17 ppb) concentrations were measured on these days (Figure 8, Table S1). Therefore, in the continental polluted air mass, the existing MCM mechanism reproduced the observed OH concentrations well. On these days, the reaction between $HO_2$ and NO was the dominant OH formation pathway (78%), followed by HONO photolysis (10%), $O_3$ photolysis (5%), the reaction between ozone and $HO_2$ (2%), and alkene ozonolysis (1%; Table 4 and Figure S3). This is similar to the findings of previous studies in the PRD conducted during autumn under polluted conditions (Tan et al., 2019). The removal of OH occurs mainly through its reaction with non-methane hydrocarbons (NMHCs; ~63%), CO (20%), $NO_2$ (10%), and $CH_4$ (4%; Table 4). Note that the reactions with NMHCs, CO, and $CH_4$ produce peroxy ($HO_2$ and $RO_2$) radicals (Stone et al., 2012). The simulated OH reactivity was $8.7 \pm 0.7$ $s^{-1}$ on average for continental air masses (Figure S4a), which is comparable to the OH reactivity measured at suburban sites (which ranged from 5 to 30 $s^{-1}$) but lower than that measured at the urban sites (which ranged from 10 to 100 $s^{-1}$; Yang et al., 2016 and references therein).

**3.2.2 Selected coastal air mass cases**

The diurnal OH patterns in the coastal air mass category (25–27 October and 5 November) were not



well reproduced by the model (Figure 7), in contrast to the continental air mass cases. The simulated results overestimated the observed OH concentration, with the daytime average $R_{S/O}$ of 1.73 (Figure 8) for these 4 case days (range 1.48 to 2.53; Figure 7). The coastal air masses showed significantly ($p <$ 0.05) lower $NO_x$ (−63%), AVOCs (−47%), BVOCs (−50%), OVOCs (−23%), and CO (−31%) concentrations compared with the continental cases (Figure 8, Table S1). The $HO_2$ and NO reaction was still the dominant source (64%) of OH in the coastal air masses, like in the continental air mass cases, but in a lower proportion than on continental days due to the lower NO concentration (Table 4 and Figure S3). The other major OH sources were HONO photolysis (15%), $O_3$ photolysis (10%), and the reaction between ozone and $HO_2$ (4%). The simulated OH reactivity was $5.4 \pm 0.33$ s$^{-1}$ on average for the coastal case (Figure S4b), which was lower than that of the continental polluted air mass ($8.7 \pm 0.7$ s$^{-1}$). As discussed below in Section 3.3, low OH reactivity could have been the cause of the model's overestimation of OH concentrations on the coastal case days. The model's overestimation of OH in coastal air masses indicates gaps in our knowledge about OH sources or sinks in relatively clean conditions with low $NO_x$ and VOCs.

### 3.2.3 The 10 October case day

During the day on 10 October, our site received continental air masses between sunrise and noon and coastal air masses between noon and sunset. This served as another case that could be used to check the model's performance on continental versus coastal air masses within the same day. On 10 October, the $R_{S/O}$ changed from 1.08 in the morning to 1.70 in the afternoon, driven by the air mass drift during continuous measurement without interruption (Figure 9). As with the continental and coastal air mass results shown above, the afternoon of 10 October showed significantly ($p < 0.05$) lower concentrations of NO (−50%), $NO_2$ (−68%), AVOCs (−42%), BVOCs (−27%), and OVOCs (−12%) compared with the morning (Table 4). With lower NO, the fraction of OH produced from $HO_2$ and NO reaction was also lower in the afternoon (54%) than in the morning (71%; Table 4 and Figure S3). Similarly, the simulated OH reactivity was lower in the afternoon ($8.1 \pm 2.0$ s$^{-1}$ on average) than in the morning (11.3 $\pm 1.1$ s$^{-1}$ on average; Figure S4c).

### 3.3 Discussion on the model–observation discrepancy



As discussed in the introduction, the model's overestimation of OH could have been caused by multiple
factors, including uncertainties in OH measurements (McKeen et al., 1997; Carslaw et al., 1999;
Mauldin et al., 2001; Mauldin et al., 2010), overestimation of OH sources ($HO_2$ and HONO; Kanaya,
et al., 2007; Zhang et al., 2006), and underestimation of OH sinks (Berresheim, 2002; Creasey et al.,
2003; Mauldin et al., 2010; Griffith et al., 2016). The possible reasons for the model's overestimation
of OH in coastal air are discussed below.
The OH measurement uncertainties in our study were described in Section 2.2. The model's
overestimation of OH in coastal air masses exceeded the measurement uncertainties (Figure 7). The
main sources of OH in the coastal air masses were the $HO_2$ + NO reaction (64%), HONO photolysis
(15%), $O_3$ photolysis (10%), and the reaction between ozone and $HO_2$ (4%). In the model, NO, HONO,
and $O_3$ were constrained by observations. To check whether the overestimation could be explained by
a larger uptake of $HO_2$ onto aerosol, we conducted a sensitivity run by increasing the aerosol uptake of
$HO_2$ (RUNγ1, Figure S5). In RUNγ1, the $HO_2$ uptake coefficient was set to unity. The simulated $HO_2$
concentration in RUNγ1 decreased by 34% compared with the base case (RUNBase). Correspondingly,
the simulated OH $R_{S/O}$ decreased to 1.42 in RUNγ1, as compared with 1.70 in RUNBase. This indicates
that the heterogeneous uptake of $HO_2$ is not sufficient to explain the OH discrepancy. A likely
explanation is the low conversion efficiency from $HO_2$ to OH at low NO concentrations, as was found
in a previous study (Sommariva et al., 2004).
We propose that the model's overestimation of OH could have been caused by unmeasured species that
were not included in the model as OH sinks. To account for these OH sinks and to investigate which
factors were important in relation to these sinks, we added a "fake" reaction into the model with the
reactivity of $k_{miss}$ ($s^{-1}$) and assumed that the reaction product would not participate in further reactions.
Assuming a pseudo-steady state of OH during the daytime (P = $k$[OH]), $k_{miss}$ was calculated as follows:

$$k_{miss} = \frac{P_{constrain}}{[OH_{obs}]} - \frac{P_{constrain}}{[OH_{sim}]} \text{ (E9)}$$

where $P_{constrain}$ is the model's calculated OH production rates, with OH constrained by observations;
$[OH_{obs}]$ is the observed OH concentration; and $[OH_{sim}]$ is the OH concentration simulated in
RUNBase. After introducing the OH removal reaction with $k_{miss}$ into the model (RUNKmiss, Figure S6),



the model better reproduced the observed OH concentrations on the coastal case days, with daytime
$R_{S/O}$ close to unity (Figure S6). This supported our estimate of $k_{miss}$. The average daytime $k_{miss}$ for the
coastal cases was $4.2 \pm 2.2$ s$^{-1}$, which corresponds to 43% of the total calculated reactivity.
We also further explored the dependence of $k_{miss}$ on different trace gases. Figure 10a shows the
correlation between $k_{miss}$ and NO concentration for the nine case days (including 10 October) between
09:00 and 15:00. At NO > 0.5 ppb, $k_{miss}$ is close to zero. At NO < 0.5 ppb, $k_{miss}$ tended to increase with
decreasing NO. Similarly, $k_{miss}$ approached zero at high concentrations of $NO_2$ (> 2.5 ppb), TEXs (>
0.25 ppb), and AVOCs (> 5 ppb; Figure 10) and increased with decreasing concentrations of $NO_2$, TEXs,
and AVOCs. High $k_{miss}$ also typically occurred at low toluene/benzene ratios and low $C_2H_2$/CO ratios,
which are indicators of an aged air mass (Figure 10; Xiao et al., 2007; Kuyper et al., 2020) Therefore,
our results suggested that the aged coastal air masses could have contained unmeasured species such as
oxygenated organic molecules (OOMs; Nie et al., 2022) and ocean-emitted gases (Thames et al., 2020)
that contributed to the missing OH reactivity, causing the model to overestimate OH concentrations on
the coastal case days.
**Summary and conclusion**
In this study, we measured OH concentrations using CIMS at a coastal site in Hong Kong in autumn
2020 to gain insights into the atmospheric oxidative capacity and to evaluate the performance of a box
model in the coastal atmosphere. The daily maximum OH concentration ranged from 2.1 to $15.4 \times 10^6$
cm$^{-3}$ over the whole campaign, with an average of $4.9 \pm 2.1 \times 10^6$ cm$^{-3}$. The air masses were categorised
into two groups based on their backward air trajectories: (1) continental masses, which contained high
concentrations of $NO_x$ and VOCs, and (2) coastal masses, which contained low concentrations of $NO_x$
and VOCs. The observed OH concentration in the continental air parcels was on average 52% higher
than in the coastal air parcels. The F0AM box model with comprehensive observational constraints
generally reproduced the observed OH in the continental cases during the daytime, with a
simulated/observed OH ratio ($R_{S/O}$) of 0.97 on average. However, the model significantly overestimated
OH concentrations in the coastal cases, with an $R_{S/O}$ of 1.70 on average during the daytime. We
attributed this overestimation to a missing OH reactivity in the aged coastal air parcels that was not



accounted for in the model. The missing OH reactivity was estimated at $4.2 \pm 2.2$ s$^{-1}$ on average between
09:00 and 15:00 and was especially larger under low NO$_x$, low AVOCs, and aged air conditions. This
result suggests that unknown products from AVOC oxidation or unknown OH-reacting gases emitted
from oceans could contribute to the missing OH reactivity in aged coastal air masses. The missing OH
reactivity in the model could cause an overestimation of the formation of secondary aerosols, such as
sulfate and nitrate, while the impacts would be even more complicated if the missing chemical species
participated in ozone formation. Further studies are necessary to pin down the exact cause(s) of the OH
overestimation, for example, through measurements of other VOC oxidation products and ocean-
emitted trace gases.
**Data availability.** All of the data used to produce this paper can be obtained by contacting Tao Wang
(two.wang@polyu.edu.hk).
**Supplement.** The online supplement for this article is available at:
**Author contributions.** TW initially conceived of the project. TW and ZW planned and organised the
overall field campaign at Hok Tsui. ZZ conducted the OH measurements using CIMS, with
contributions from TW and ZW. YC performed the aerosol size distribution measurements. YQ
performed the OVOC measurements using PTR-MS. MX and YC performed the HONO measurements
using ToF-CIMS. YW assisted with HONO calibration. ZZ performed the box model analysis and
sensitivity test with contributions from EX and QC. ZZ, TW, and QC analysed the data and interpreted
the results, with contributions from MX. ZZ, TW and QC wrote the paper. All of the authors reviewed
and commented on the paper.
**Competing interests.** One author (Tao Wang) is a member of the editorial board of Atmospheric
Chemistry and Physics. The peer-review process was guided by an independent editor, and the authors
have no other competing interests to declare.
**Acknowledgments**
We thank David Tanner, Dr Wei Pu, and Dr Weihao Wang for developing the OH CIMS. We are also
grateful to the Environmental Protection Department of Hong Kong, for loaning the CIMS instrument
and providing access to its station and data on trace gases, and to the Hong Kong Observatory, for
supplying meteorology data.



1   **Financial support.** This research was financially supported by the Hong Kong Research Grants

2   Council (T24-504/17-N and 15223221).



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



**Figures and Tables**

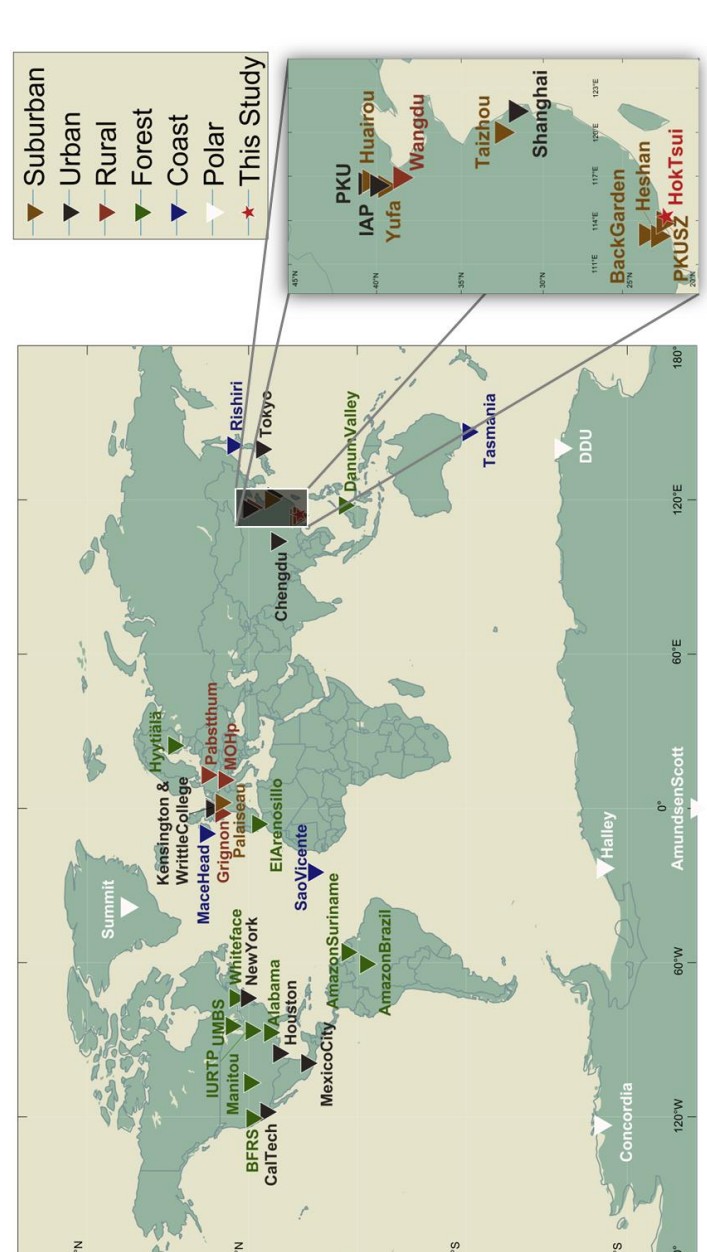

Figure 1. OH measurements gathered around the world to date.

Table 1. Summary of studies reporting OH and HO$_2$ measurements and comparing them with model predictions (refer to Figure 1 for site locations)



| Comparison results (OH only) | Reference | Time | Location in Figure 1 | Site types | Measurement notes | OH conc. $10^6$ cm$^{-3}$ | HO$_2$ conc. $10^8$ cm$^{-3}$ | Ratio notes | OH R$_{SO}$ | HO$_2$ R$_{SO}$ | Other references targeting the same site |
|---|---|---|---|---|---|---|---|---|---|---|---|
| Overprediction | (Berresheim, 2002); (Sommariva et al., 2004); | June–July 1999 | MaceHead | Coast | Mean (All) | 0.12 | N/A | Mean (17 June, Coastal) | 2 | N/A | (Carslaw et al., 1999); (Berresheim et al., 2013), 2014 |
| | | | | | Peaks (Mean Clean) | 2.5 | | Mean (30 July, Continental) | ~1 | N/A | |
| | | | | | Peaks (Pollution) | 18 & 12 | | | | | |
| Overprediction | (Creasey et al., 2003); | Jan–Feb 1999 | Tasmania | Coast | Mean (Peaks) | 3.5 | 2 | Mean (7–8 Feb) | ~1.11 | N/A | N/A |
| | | | | | Peaks (Range) | 2 to 5.5 | 1 to 2.5 | Mean (15–16 Feb) | ~1.32 | ~2 | |
| Overprediction | (Kanaya et al., 2007a); | September 2003 | Rishiri | Island Coast | Peaks (Mean) | 2.7 | 1.45 | OH rectified by constrained HO$_2$ | 1.35 | 1.89 | N/A |
| Overprediction | Mauldin et al., 2010 | Nov–Jan 2003-04 | AmundsenScot | Antarctica | Mean (Range) | 1.5 to 2.5 | N/A | Mean | ~2 | N/A | Mauldin, Eisele, Kosciuch, et al., 2001 |
| Overprediction | (Kukui et al., 2014) | Dec–Jan 2011-2012 | Concordia | Antarctica | Mean (All) | 3.1 | 0.99* | Mean (w/PSS HONO) | 0.72 | 1.02* | N/A |
| | | | | | Peaks (Mean) | 5.2 | 1.7* | Mean (w/measured HONO) | 2.19 | 1.84* | |
| | | | | | Mean (range) | 0.3 to 7.5 | 0.11 to 2* | Mean (13:00 w/o glyoxal) | 2.4 | 1.5 | |
| Overprediction | (Dusanter et al., 2009a); a(O), b (M) | March 2006 | MexicoCity | Urban | Peaks (Range) | 2 to 15 | 0.56 to 4.5 | Mean (Morning, polluted) | ~0.5 to ~1 | 0.2 to ~1 | N/A |
| | | | | | Peaks (Mean) | 4.6 | 1.9 | Mean (11:00–14:30) | 1.7 | ~1 | |
| | | | | | | | | Mean (After 14:30) | ~1 | ~1 | |
| Overprediction | (Bloss et al., 2007) | Jan–Feb 2005 | Halley | Polar | Mean (All) | 0.39 | 2.04 | Peak (S1: Conventional) | 0.67 | N/A | N/A |
| | | | | | Peak (Mean) | 0.79 | 4.03 | Peak (S2: S1 + halogen oxides) | 1.64 | | |
| | | | | | | | | Peak (S3: S2 + possible VOCs) | 1.27 | | |
| Overprediction | (Holland et al., 2003) | Jul–Aug 1998 | Pabstthum | Rural | Peaks (Range) | 6 to 8 | 5 to 7.4 | Mean (Low NO$_x$) | 2 | 1.4 | N/A |
| Overprediction | (Whalley et al., 2018) | Jul–Aug 2012 | Kensington | Urban | Mean (Noon; S-W air) | ~2.2 | ~0.2 | Mean (Air mass: South-westerly) | 1.25 | ~4 | N/A |
| | | | | | Mean (Noon; E, polluted air) | ~3 | ~0.5 | Mean (Air mass: Easterly, polluted) | 2 | 10 | |
| Overprediction | (Griffith et al., 2016) | May–June 2010 | CalNexLA | Urban | Peaks (Range) | 1.5 to 9 | 0.8* to 10* | Mean (Weekend) | 1.43 | 0.77* | (Volz-Thomas et al., 2003b); (Volz-Thomas et al., 2003a) |
| | | | | | Peaks (Weekdays) | ~4 | ~3* | Mean (Weekday) | 1 | 0.33* | |
| | | | | | Peaks (Weekend) | ~5 | ~8* | | | | |





| Comparison results (OH only) | Reference | Time | Location in Figure 1 | Site types | Measurement notes | OH conc. $10^6$ cm$^{-3}$ | HO2 conc. $10^8$ cm$^{-3}$ | Ratio notes | OH $R_{SO}$ | HO2 $R_{SO}$ | Other references targeting the same site |
|---|---|---|---|---|---|---|---|---|---|---|---|
| Underprediction | (Hofzumahaus et al., 2009); (Lu et al., 2012) | July 2006 | BackGarden | Rural | Peaks (Mean) | 15 | 15 | Mean (Range, NO < 1 ppb) | 0.2 to 0.33 | N/A | N/A |
| | | | | | | | | Mean (Lu et al., 2012) | 0.5 | | |
| Underprediction | (Whalley et al., 2011) | Apr–May 2008 | DanumValley | Rainforest | Peaks (Mean) | 2.5 | 3 | w/C3H8 recycling mechanism (Peeters et al., 2009) | -0.63 | -0.5 | (M. Pugh et al., 2010) |
| | | | | | | | | | 0.72 | | |
| Underprediction | (Liao et al., 2011) | May–June 2007; Jun–Jul 2008 | Summit | Polar | Mean (2007 spring) | 3.0 | 2.7* | 2007 spring w/o BrO & w BrO | 0.78 | 0.87 | (Sjostedt et al., 2007) |
| | | | | | Mean (2008 summer) | 4.1 | 4.2* | 2008 summer w/o BrO & w BrO | 0.54 | 0.96 | |
| | | | | | | | | | 0.56 | | |
| Underprediction | (Wolfe et al., 2014) | Aug 2010 | Manitou | Forest | Peaks (Range) | 3 to 10 | 24.6 to 44.3 | Peak | -0.625 | 0.33 | (Kim et al., 2013) |
| Underprediction | (Tan et al., 2019) | Oct–Nov 2014; Jul–Aug | Heshan | Suburban | Peaks (Mean) | 4.5 | 3 | Budget analysis only | N/A | 1 | N/A |
| | | | | | | | | Mean (2008) | -0.4 | -0.57 | |
| Underprediction | (Griffith et al., 2013) | Jul–Aug 2008; Jul–Aug 2009 | UMBS | Forest | Peak (Mean, 2008) | ~3.3 | ~7 | Mean (2008 w/ISOP mechanisms) | | -1.3 | N/A |
| | | | | | Peak (Mean, 2009) | ~1.6 | ~4.8 | Mean (2009) | 0.9 | -0.6 | |
| | | | | | | | | Mean (2009 w/ISOP mechanisms) | | -1.5 | |
| Underprediction | K.D. Lu et al., 2013 | Sept 2006 | Yufa | Urban | Peaks (Range) | 4 to 17 | 2 to 24 | Mean (NO <0.1 ppb) | 0.38 | -1 | N/A |
| | | | | | Peaks (Mean) | ~7 | ~1.5 | Mean (NO > 1 ppb) | -1 | -1 | |
| Underprediction | (Mao et al., 2012) | Jun–Jul 2009 | BFRS | Forest | Peaks (Mean w/ interference) | ~4.5 | N/A | Mean (w/ interference) | 0.32 | N/A | N/A |
| | | | | | Peaks (Mean w/o interference) | ~1.8 | | Mean (w/o interference) | 0.71 | | |
| Underprediction | (Tan et al., 2017); (Lu et al., 2019b) | Summer 2014 | Wangdu | Rural | Peaks (Range noontime) | 5 to 15 | 3 to 14 | Mean (NO > 0.3 ppb) | -1 | 10 (NO > 4 ppb) | N/A |
| | | | | | | | | Mean (NO <0.3 ppb, afternoon) | 0.5 | N/A | |
| Underprediction | (Lew et al., 2020) | Jul 2015 | IURTP | Forest | Peaks (Mean) | 4 | 10 | Mean (Daytime) | 0.83 | 1.10 to 1.32 | N/A |
| | | | | | | | | Mean (Evening and morning) | 0.50 | | |
| Underprediction | (Lelieveld et al., 2008) | Oct 2005 | AmazonSuriname | Forest (Flight) | Mean (Forest boundary) | 5.6 | 10.5 | Mean (w/MIM: Mainz ISOP mechanism) | 0.1 to 0.2 | N/A | N/A |
| | | | | | Mean (Forest free troposphere) | 8.2 | 4.9 | Mean (w/MIM2+: extra 40% to 80% OH recycle) | -1 | | |
| | | | | | Mean (Atlantic boundary) | 9.0 | 6.7 | | | | |
| | | | | | Mean (Atlantic free troposphere) | 10.1 | 5.5 | | | | |




| Comparison results (OH only) | Reference | Time | Location in Figure 1 | Site types | Measurement notes | OH conc. $10^6$ cm$^{-3}$ | HO$_2$ conc. $10^8$ cm$^{-3}$ | Ratio notes | OH R$_{XO}$ | HO$_2$ R$_{XO}$ | Other reference targeting the same site |
|---|---|---|---|---|---|---|---|---|---|---|---|
| Good match | (Ren et al., 2003a; a(O), b(M)) | Jun–Aug 2001 | NewYork | Urban | Peaks (Range) / Peaks (Mean) | 2 to 20 / 7 | 0.5 to 6 / 1 | Mean | 0.91 | 0.81 | N/A |
| Good match | (Ren et al., 2006) | Jul–Aug 2002 | Whiteface | Forest | Peaks (Mean) | 2.6 | 4.9 | Mean | 1.22 | 0.83 | N/A |
| Good match | (Kanaya et al., 2007b) | Jan-Feb & Jul–Aug 2004 | Tokyo | Urban | Peaks (Mean, winter) / Peaks (Mean, summer) | 1.5 / 6.3 | 0.27 / 1.4 | Peaks (Mean, winter) / Peaks (Mean, summer) | 0.99 / 0.81 | 0.71 / 1.22 | N/A |
| Good match | (Feiner et al., 2016; Kaiser et al., 2016) | Jun–Jul 2013 | Alabama | Forest | Peaks (Mean) | 1 | 6.64 | Peaks (Mean) | ~1 | ~1 | N/A |
| Good match | (Jeong et al., 2022) | Feb–Mar 2014 | AmazonBrazil | Forest | Peaks (Mean 10:00-15:00) / Peaks (Range) | 1 / ~1 to ~2.8 | N/A | Mean | 1 | N/A | N/A |
| Good match | (Hens et al., 2014) | Summer 2010 | Hyytiälä | Forest | Mean (Above-Canopy) / Mean (Ground) | 3.5 / ~1.8 to ~1.2 | N/A | Mean | 1 | 0.3 | (Petäjä et al., 2009); (Novelli et al., 2014b) |
| Good match | (Emmerson et al., 2007) | Jul–Aug 2003 | WrittleCollege | Urban | Peaks (Range) | 1.2 to 7.5 | 0.16 to 3.3 | Mean | 1.24 | 1.07 | N/A |
| Good match | (Ren et al., 2013) | Apr–May 2009 | Houston | Urban | Peak (Mean) | ~8.8 | ~6.2 | Mean | 0.9 | 1.22 | (Mao et al., 2010); (Chen et al., 2010) |
| Good match | (Ma et al., 2019) | Nov–Dec 2017 | PKU | Urban | Peaks (Mean clean) / Peaks (Mean polluted) | 2 / 1.5 | 0.4 / 0.3 | Mean (clean) / Mean (pollued) | ~1 / ~0.66 | ~0.66 / 0.08 | N/A |
| Good match with missing source | (Whalley et al., 2021) | Summer 2017 | IAP | Urban | Peak (All) | 28 | 10 | Mean (NO < 1 ppb) | ~1 | 1.83 | (Slater et al., 2020) |
| Good match with underpredicted HO$_2$ | (Zhang et al., 2022b) | Nov–Dec 2019 | Shanghai | Urban | Peaks (Mean) | 2.7 | 0.8 | N/A | N/A | N/A | N/A |
| No comparison | (Kukui et al., 2008) | June–July 2007 | Grignon | Suburban | Peak (July 6) | ~23 | ~2 | N/A | N/A | N/A | N/A |
| No comparison | (Wang et al., 2021) | Oct–Nov 2018 | PKUSZ | Suburban | Peaks (Mean) | 5.3 | 4.2 | N/A | N/A | N/A | (Wang et al., 2019a) |
| No comparison | (Rohrer and Berresheim, 2006) | 1999–2003 | MOHp | Rural | Mean (All) | 1.97 | N/A | N/A | N/A | N/A | (Handisides et al., 2003); (Novelli et al., 2014b) |





| No comparison | (Zhang et al., 2022a) | Aug.–Sept 2019 | Chengdu | Urban | Peaks (Range, PKU-LIF) Peaks (Range, AIOFM-LIF) | 1.6 to 15 2.1 to 15.9 | N/A N/A | N/A | N/A | N/A | N/A |

**Notes**

ISOP:     Isoprene

AIOFM:     Laser- induced fluorescence instrument by the Anhui Institute of Optics Fine Mechanics, Chinese Academy of Sciences

Mean:     Campaign average concentration or ratio

Peak:     Campaign maximum concentration or ratio

Peaks (Mean):     Maximum concentration or ratio for the averaged diurnal or averaged cases

Mean (Range):     Daily average concentration or ratio range for the campaign or cases

Peaks (Range):     Maximum concentration or ratio range for the campaign or cases

w/ and w/o:     Considered or did not consider the specific mechanism, species, or interference

~:     The result is based on the figure or description, and the exact number is not mentioned in the article

N/A:     Not available in the article

*:     The $HO_2$ result includes some $RO_2$ species.



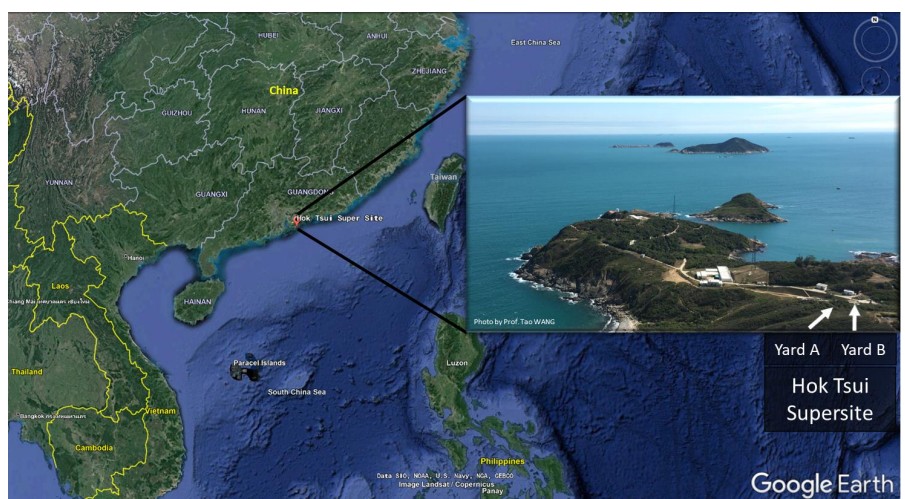

Figure 2. The location of the Hok Tsui Air Monitoring Supersite in Hong Kong, South China.
The map is from © Google Earth.



Table 2. Measuring instruments and measured species in the field campaign

| Species | Instruments | Time resolution | Detection limit |
|---|---|---|---|
| OH | Nitrate-quadrupole chemical ionization mass spectrometer (CIMS) | 10 s | Lab: $1.5 \times 10^5$ cm$^{-3}$ Daytime: $1 \times 10^6$ cm$^{-3}$ |
| NO, NO$_2$ | Chemiluminescence/photolytic converter (Thermo, Model 42i) | 1 min | 0.06 ppb |
| Ozone | Ozone analyser, model 49i, Thermo Scientific | 1 min | 0.5 ppb |
| $J_{NO2}$ | Filter radiometer, Metcon | 1 min | $4 \times 10^{-5}$ s$^{-1}$ |
| HONO | Iodide-ToF-CIMS, Aerodyne Inc | 1 s | 0.2 ppt |
| Particle number size distribution | Scanning mobility particle sizer (SMPS), TSI | 5 mins | 1 particle cm$^{-3}$ |
| VOCs | Gas chromatograph system (GC-MS/FID; GC955 Series 611/811, Syntech Spectras) | 1 hour | ~10 ppt |
| | Proton-transfer-reaction mass spectrometry (PTR-MS; PTR-QMS 500, IONICON Analytik, Austria) | 5 mins | 20 ppt |
| OVOCs | High-performance liquid chromatography (HPLC); PTR-ToF-MS, IONICON Analytic; | 1 s | ~10 ppt |



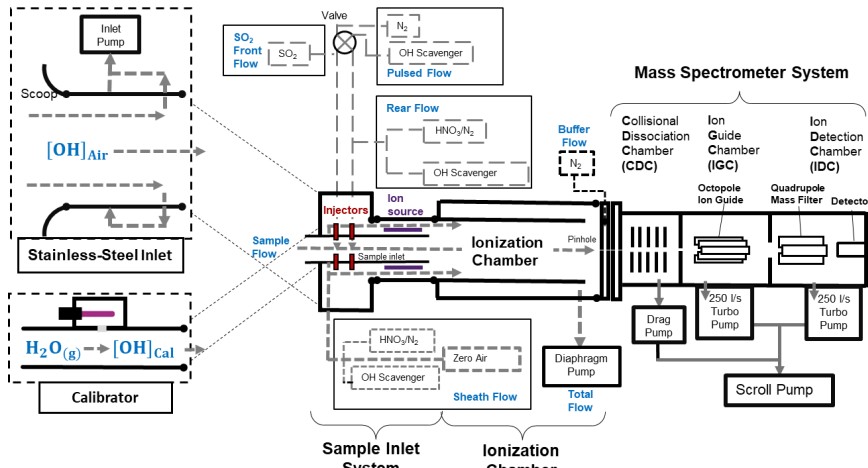

Figure 3. Schematics of the CIMS system which consists of a stainless-steel inlet, a sample inlet, an ionization chamber, a mass spectrometer system, and a calibration unit. The CIMS measures the ambient OH concentration when connecting to the stainless-steel inlet whereas, during calibration, the calibration unit is connected to the CIMS instead. Details of the CIMS setup and calibration can be found in Section 2.2.




Table 3. Technical specifications of the OH-CIMS

| Flow position | Gas | Flow rates (sccm) | Specification for measurement | |
|---|---|---|---|---|
| **Front injectors** | $SO_2$ (0.9%) | 5 | Sample flow [$SO_2$] (ppm) | 12 |
| | NO (0.9%) | 0 | Sample flow [NO] (ppm) | 0 |
| **Pulse valve** | $C_3F_6$ (99.9%) | 2 | Elimination rate | 95% |
| | $N_2$ | 2 | Switching time (min) | 2 |
| **Rear injectors** | $C_3F_6$ (99.9%) | 2 | Reaction time (ms) | 47 |
| | HNO3 | 10 | Sample flow [$C_3F_6$] (ppm) | 1072 |
| **Sheath flow** | $C_3F_6$ (99.9%) | 2 | Sheath flow [$C_3F_6$] (ppm) | 159 |
| | $HNO_3$ | 10 | Detection Limit in lab ($\times 10^5$ cm$^{-3}$) | 1.5 |
| | Zero air | 126,000 | Sheath flow speed (cm/s) | 25 |
| **Buffer** | $N_2$ | 440 | Reynolds number in ionization chamber | > 4000 |
| **Total flow** | | 168,000 | Sample flow speed (cm/s) | 55 |
| **Sample flow** | | 3727 | Stainless-steel inlet flow speed (m/s) | 5 |
| **Calibration flow** | | 10,000 | Calibration flow speed (cm/s) | 65 |
| **Calibration factors** for OH with N$^{18}$O$_3^-$ as the reagent ion $\times 10^{-8}$ ((OH·cm$^{-3}$)/Hz) | | | | 1.21 |
| **Detection limit** for OH measurement over the whole campaign ($\times 10^5$ cm$^{-3}$) (2σ) | | | Daytime | 10 |
| | | | Nighttime | 7.7 |
| **Uncertainties** for OH measurement | | | | 44% |



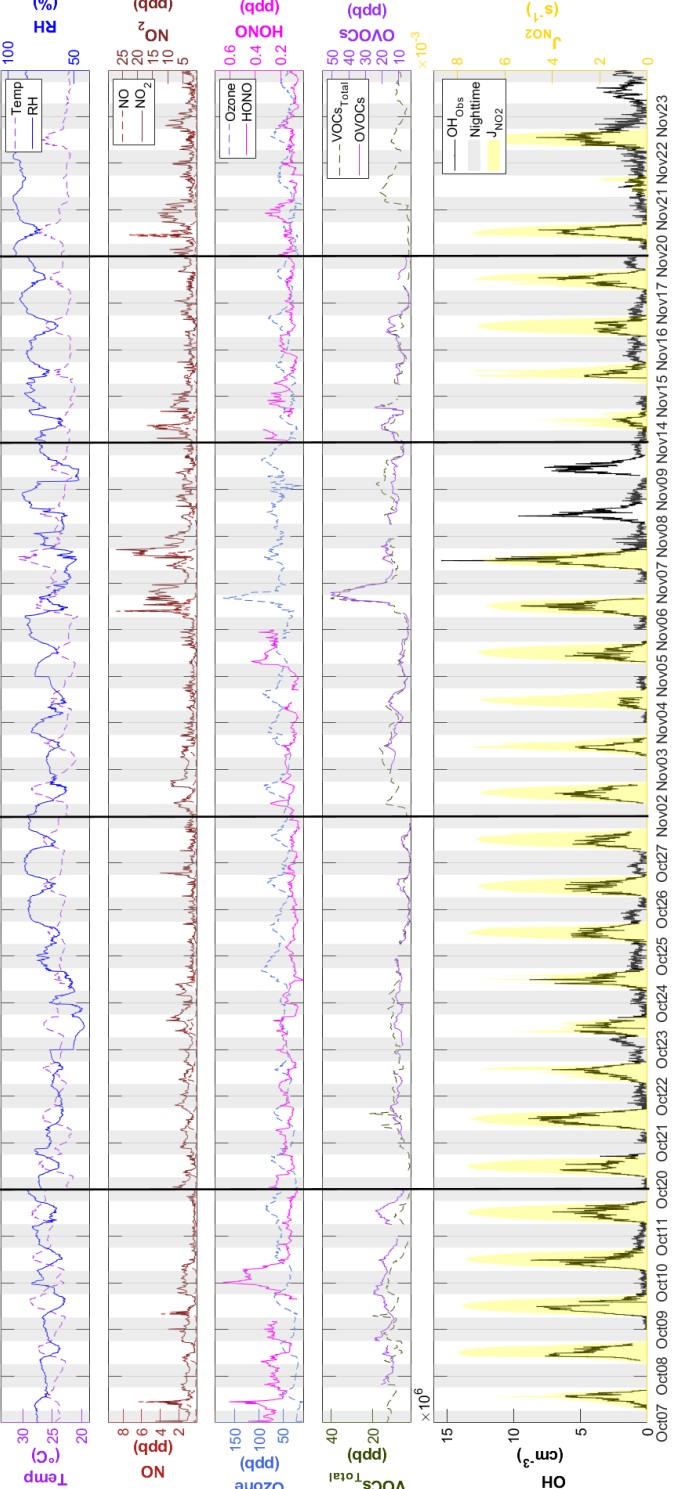

Figure 4. Time series of OH between 7 October and 23 November with measured weather conditions (temperature and RH), OH primary sources (ozone and

HONO), $NO_x$ (NO and $NO_2$), organic compounds (total VOCs and OVOCs), and photolysis frequency ($J_{NO2}$). All measurement data shown are 10 min averages.

The black lines separate the non-continuous days during measurement. The grey shaded area denotes nighttime.

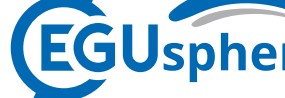

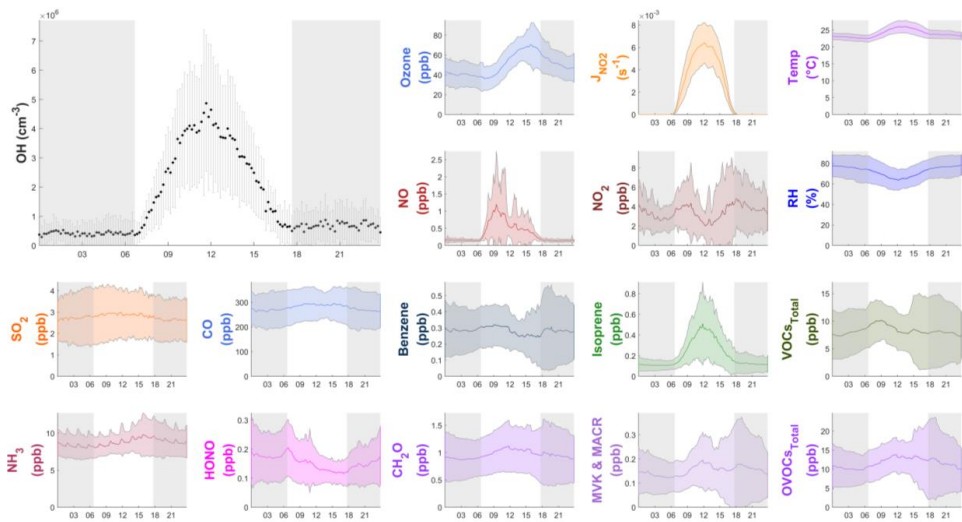

Figure 5. Diurnal profiles of the average (±1σ) concentrations of OH, other chemical species, and meteorological parameters ($T$, RH, $J_{NO2}$) during the field campaign. The grey shaded area denotes nighttime.

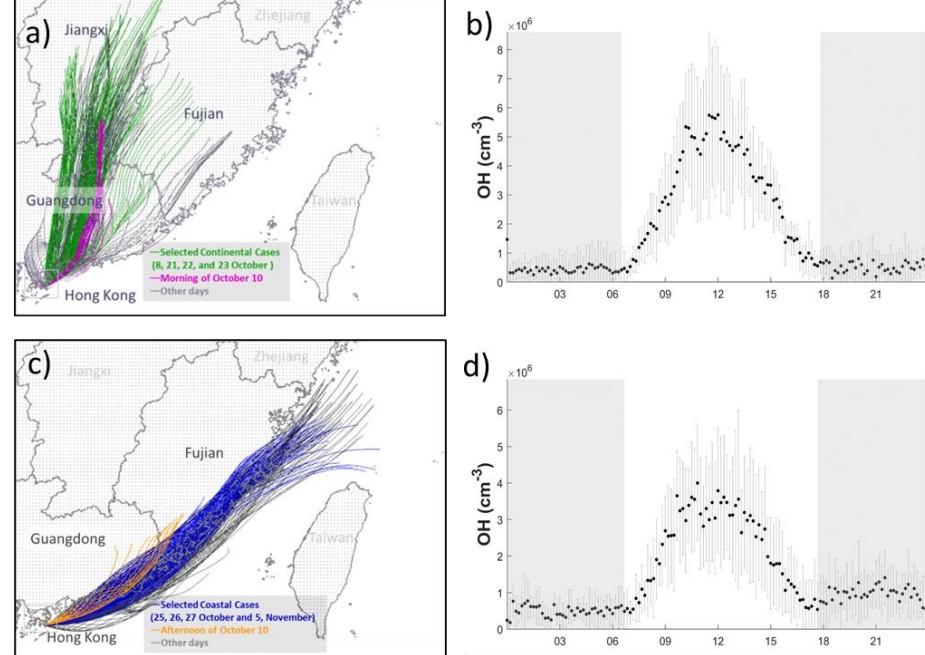

Figure 6. 24 h back trajectories of the continental (a) and coastal (c) cases over the whole measurement period. The selected days for coastal, continental, and mixed cases are labelled in different colours. (b)




and (d) show the average concentration of OH with standard deviation in continental and coastal air
masses, respectively.

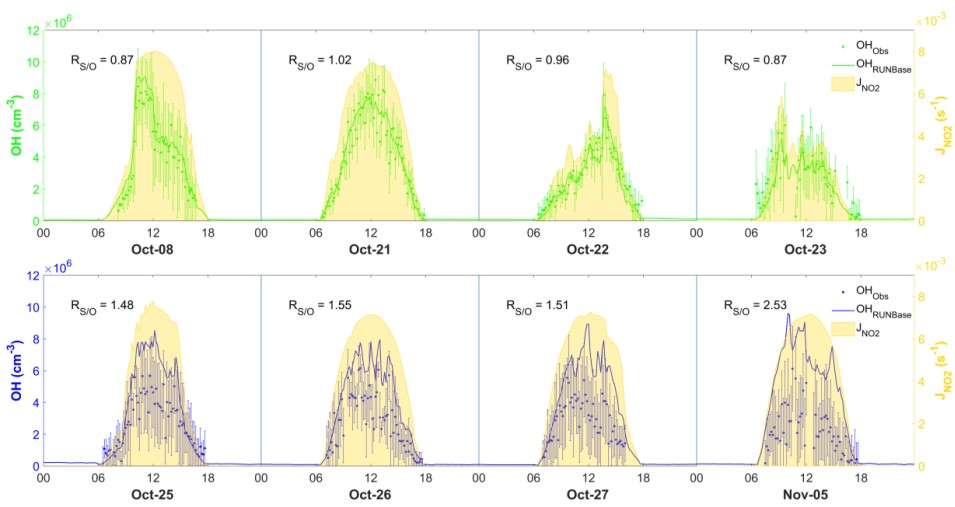

Figure 7. Comparison between observed (dots) and simulated (lines) OH in the four continental cases
(top panel) and the four coastal cases (lower panel), also showing measurement uncertainty (error bars)
and $J_{NO2}$ measurement (yellow shades).

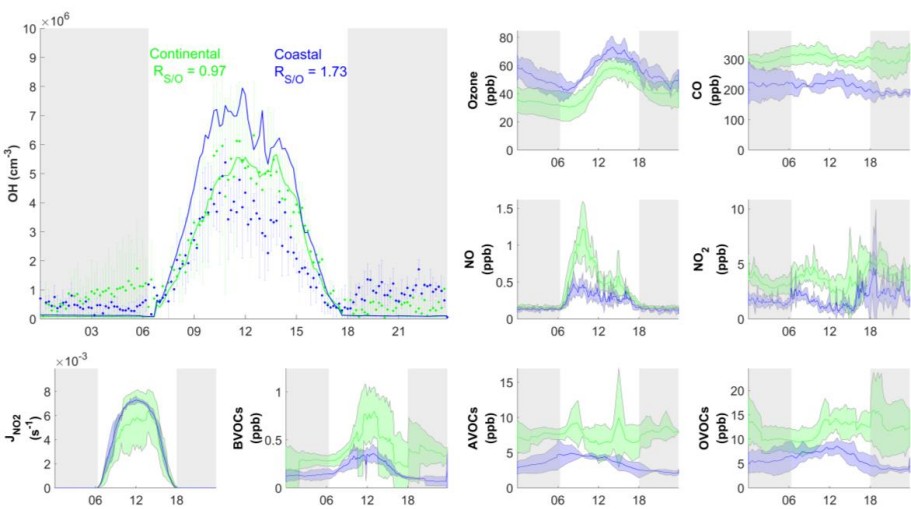

Figure 8. Diurnal profiles of average (± 1σ) concentrations of measured (dots) and simulated (RUNBase,
line) OH concentration and important trace gases for selected cases in continental (green) and coastal
(blue) air masses. The grey shaded area denotes nighttime.





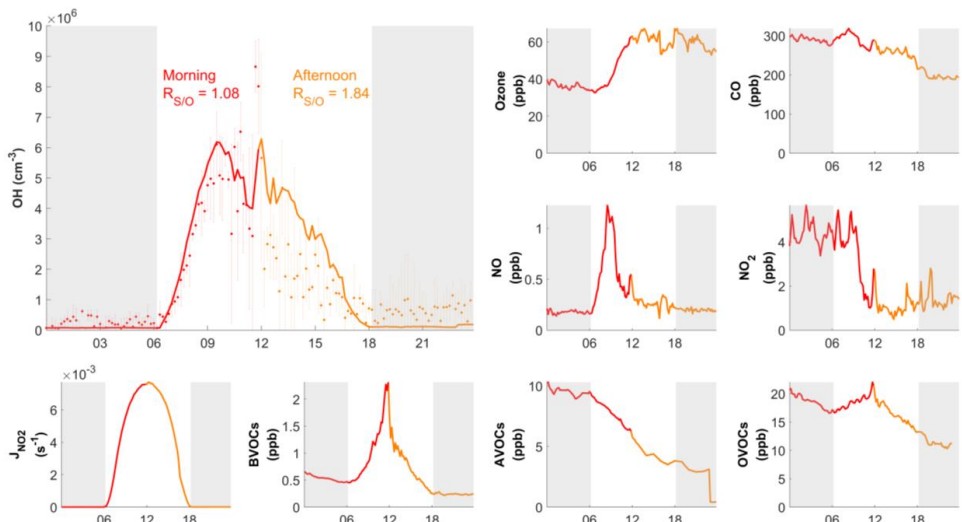

Figure 9. Diurnal profiles of measured (dots) and simulated (RUNBase, line) OH on 10 October 2020,

with other chemical species. The air mass drifted from continental (red) in the morning to coastal

(orange) in the afternoon. The grey shaded area denotes nighttime.





1    Table 4. OH budgets for the selected continental and coastal cases, morning and afternoon of 10

2    October.

| Continental cases | | Coastal cases | | 10 October (morning) | | 10 October (afternoon) | |
|---|---|---|---|---|---|---|---|
| Production | | | | | | | |
| $HO_2 + NO$ | 77.57% | $HO_2 + NO$ | 63.87% | $HO_2 + NO$ | 70.88% | $HO_2 + NO$ | 53.73% |
| $HONO + hv$ | 10.09% | $HONO + hv$ | 15.01% | $HONO + hv$ | 12.19% | $HONO + hv$ | 14.14% |
| $O^1D$ | 4.82% | $O^1D$ | 10.14% | $O^1D$ | 8.39% | $O^1D$ | 13.68% |
| $HO_2 + O_3$ | 1.79% | $HO_2 + O_3$ | 4.32% | $HO_2 + O_3$ | 1.94% | $HO_2 + O_3$ | 5.65% |
| $CH_3CCH_3OOC$ | 1.25% | $CH_3CCH_3OOB$ | 1.22% | $H_2O_2 + hv$ | 0.68% | $H_2O_2 + hv$ | 2.56% |
| Other | 4.49% | Other | 5.43% | Other | 5.93% | Other | 10.23% |
| Loss | | | | | | | |
| CO | 20.04% | CO | 22.75% | $C_5H_8$ | 15.98% | CO | 15.40% |
| $NO_2$ | 9.81% | $C_5H_8$ | 7.64% | CO | 13.82% | $C_5H_8$ | 12.92% |
| $C_5H_8$ | 8.77% | $C_2H_5CHO$ | 7.29% | $C_2H_5CHO$ | 8.56% | $C_2H_5CHO$ | 10.96% |
| $C_2H_5CHO$ | 7.91% | $NO_2$ | 7.20% | $CH_3CHO$ | 8.51% | $CH_3CHO$ | 7.31% |
| $CH_3CHO$ | 7.73% | $CH_3CHO$ | 6.79% | $NO_2$ | 6.00% | $CH_4$ | 3.50% |
| $CH_4$ | 3.68% | $CH_4$ | 5.77% | HCHO | 3.08% | HCHO | 3.49% |
| HCHO | 2.76% | HCHO | 2.45% | $CH_4$ | 2.73% | $NO_2$ | 2.75% |
| $O_3$ | 1.38% | $O_3$ | 1.99% | ACR | 1.63% | ACR | 1.69% |
| $H_2$ | 1.26% | $H_2$ | 1.67% | $HOCH_2CHO$ | 1.42% | $HOCH_2CHO$ | 1.51% |
| Other | 36.67% | Other | 36.45% | Other | 38.28% | Other | 40.47% |

3    Notes: ACR- acrolein



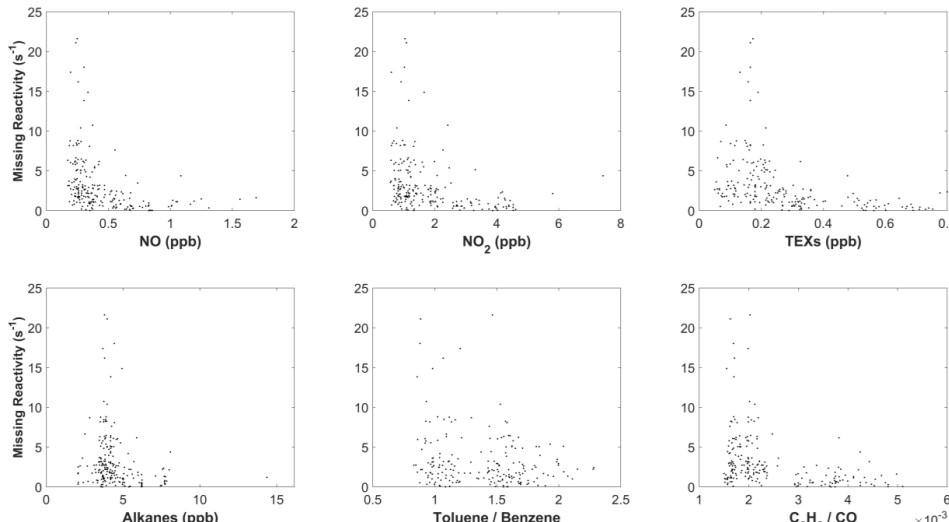

Figure 10. The dependence of calculated missing reactivity on a) NO, b) $NO_2$, c) TEXs (toluene,

ethylbenzene, and xylene), d) alkanes ($C_2$ to $C_8$), e) the ratio of toluene to benzene, and f) the ratio of

$C_2H_2$ to CO.

