# Peer review of "OH measurements in the coastal atmosphere of South China: Possible missing OH sinks in aged"

_EGUsphere, 2022_

## Author Comment (AC1)

Thank you for your very detailed and helpful comments on our manuscript (EGUSPHERE-2022-854). The comments lead us to re-examine the simulation and observation results and the discussions we had made. Please find our itemized responses below and the corrections in the re-submitted files. The original comments are in **black.** The responses to the comments are in **blue**. The changes in the manuscript and the supplementary were highlighted in yellow and were cited in **purple** in this response.

Comment # 1

The paper reports ground-based measurements of OH concentrations that were obtained in a coastal area in southern China in the fall of 2020. Additional measurements of trace gases ($O_3$, NO, $NO_2$, HONO, CO, VOCs, OVOCs) and meteorological parameters were used as input to a zero-dimensional box model to simulate the OH concentrations which are then compared to the observations. Modeled and measured OH concentrations agreed during the day when continental air arrived at the measurement site, but the model overestimates the observed OH concentrations in coastal air by a factor of 1.7. The disagreement is attributed to unmeasured atmospheric components which are supposed to be missing in the model as OH sinks. Atmospheric OH measurements are generally difficult and rare. New observations in previously understudied regions, as in this work, are therefore of potential interest. However, the paper in its current form has major shortcomings. In particular, since important quantities such as jO1D, $HO_2$, and OH reactivity were not measured during the field campaign, few new insights can be gained from the reported OH observations that would expand our understanding of atmospheric OH chemistry. The manuscript would potentially qualify as a Measurement Report if (a) the documentation of the measurement instrument is improved, (b) the analysis of measured OH diurnal cycles is extended to include nocturnal values, and (c) the interpretation of the comparison between model and measurement is more balanced. The title of the paper would need to be adjusted accordingly.

Response: Thanks for the comments. We agree that lacking the measurement of JO1D, $HO_2$, and OH reactivity in our field study is the major shortcoming which hinders further analysis of the data. The revised manuscript discussed these shortcomings in detail. Following referee's suggestions, we have added more descriptions of the instrument and interpretation of nighttime OH and balanced model and measurement comparison. See below for the response to the specific comment and corresponding change.

We have revised the title as "OH measurements in the coastal atmosphere of South China: Possible missing OH sinks in aged air masses".

General comments

1. Instrumental description

One problem is that the applied OH instrument is newly developed. The applied CIMS concept is well known in the literature, but the specific characterization, calibration, and treatment of potential interferences of the present instrument are not well documented. The authors refer to a preprint that was submitted to the journal AMT (Pu et al., Development of a chemical ionization mass spectrometry system for measurement of atmospheric OH radical, amt-2020-252) but was not accepted for final publication. Reference to that manuscript is problematic because it is not clear to the readers which of the statements made there are valid for the current work or led to a rejection. Without further explanation, the cited work is insufficient to support the quality of the measurements here. The present paper should stand on its own independently of the preprint in AMT.

Response: The previous preprint in AMT didn't move to formal publication because one referee thought that our CIMS did not outperform the CIMS in other groups. We agree with the current referee that it is better for this article to stand on its own. Thus, some contents in the AMTD preprint article, including the calibration, detection limit, uncertainties calculation, calibration uncertainties, and the sensitivity optimization were modified and added in the updated supplementary to support the quality of the measurement.

The contents added in the supplementary as below:

**"Calibration**
The calibration was performed by applying the calibrator shown in Figure 3 to the CIMS. The calibration flow passes through the water bubbler and carries $H_2O$. When the humid calibration flow is exposed to the Pen Ray mercury lamp (Analytik Jena, UVP Pen Ray), the OH radicals produced by $H_2O$ photolysis as shown below:
$$H_2O + hv(184.9\ nm) \rightarrow H + OH \tag{SR1}$$
The OH concentration ($[OH]$) produced by the calibrator is calculated by SE1.
$$[OH] = [H_2O] * \sigma_{H_2O} * \Phi * It \tag{SE1}$$
$[H_2O]$ is the water concentration in the calibration flow which is calculated from the temperature, saturated water vapor pressure, and relative humidity. $\sigma_{H_2O}$ (= $7.14 \times 10^{-20}$ cm$^2$; Cantrell et al., 1997) is the photolysis cross-section of water vapor, while $\Phi$ represents the photolysis quantum yield and was assumed to be 1 (Kürten et al., 2012). The photon flux ($It$) was determined using the chemical actinometry method. This method measures the mixing ratio of $N_2O$ and its photolysis products to determine the product $It$ value of the lamp in the calibrator while the $N_2O$ photolysis and $H_2O$ photolysis require the same photon intensity (184.9 nm). The reactions and equations of $It$ determination were presented by (Kürten et al., 2012). In this study, the $It$ values were measured before and after the field campaign and no significant difference was found.

By calculating the [OH]$_{CAL}$ produced by the calibrator, the signal response to OH ($TS_{97}$ - $BS_{97}$), and reagent ion ($S_{62}$), the calibration factor ($C$) can be calculated by following the equation which was transformed from E1.

$$C = \frac{1}{[OH]_{CAL}} \times \frac{TS_{97}-BS_{97}}{S_{64}} \tag{SE2}$$

**Detection limit**
The detection limit can be calculated as follows,

$$DL = \frac{1}{C} \times \frac{n * \sigma_{BS97}}{S_{64}} \tag{SE3}$$

Where DL is the detection limit in $10^6$ molecule/cm$^3$, $C$ is the calibration factor, and $n$ is the ratio of signal to noise S/N. $\sigma_{BS97}$ represents the standard deviation of the signal intensity of $HSO_4^-$ at 97 m/z when the scavenger was added through front injectors, and $S_{64}$ represents the signal intensity of $NO_3^-$ at 64 m/z. The detection limit (S/N=2, average time=6 minutes) in the laboratory was 0.15 $*10^6$ molecule cm$^{-3}$.

**Uncertainty calculation**
The uncertainty of OH measurements is calculated by the rules for propagation for the uncertainty.

Propagation Rules for Addition ($y = x_1 + x_2$):     $e_y = \sqrt{e_{x1}^2 + e_{x2}^2}$     (SE4)

Propagation Rules for Multiplication ($y = x_1 \times x_2$):     $\frac{e_y}{y} = \sqrt{\left(\frac{e_{x1}}{x_1}\right)^2 + \left(\frac{e_{x2}}{x_2}\right)^2}$ (SE5)

Where $e_y$, $e_{x1}$, $e_{x2}$, are the absolute uncertainty for y, $x_1$, and $x_2$.

**Calibration Uncertainties**
The calibration uncertainty is calculated by the uncertainties of all the parameters involved in the SE2 which includes the uncertainty of calculated OH radicals concentration and the precision of the measurements of signal at 64 m/z and 97 m/z. Due to the equation SE1, the uncertainty of OH radical ([OH]$_{CAL}$) will further be contributed by the uncertainty of $It$ ~36%, $\sigma_{H_2O}$ (~5%), $\phi_{H_2O}$ (<1%, Cantrell et al. 1997), and calculated water concentration based on the measured Temperature and relative humidity (~10%). The precision of the measurements signal at 64 m/z and 97 m/z ($\frac{TS_{97}-BS_{97}}{S_{64}}$) of the CIMS instrument (2σ) during calibration was 11% (for 6 min integration time). Considering all the above uncertainties and calculated by the rules (SE3 and SE4), the overall uncertainty for the calibration factor can be calculated by the well-known uncertainty formula. The uncertainty for the calibration factor was about 38% in this study.

**Sensitivity optimization**
The sensitivity (S) of the CIMS instrument to the OH radicals depends on the conversion efficiency of OH to H$_2$SO$_4$ in the chemical conversion region (E$_{Conv}$), the ionization efficiency of H$_2$SO$_4$ to $HSO_4^-$ in chemical ionization region (E$_{Ion}$), and the

ion-transmitted efficiency of $HSO_4^-$ from sample inlet to mass spectrometer system ($E_{Trans}$):

$$S \sim E_{Conv} \cdot E_{Ion} \cdot E_{Trans}$$

$E_{Conv}$ is dependent on the reaction time and the $SO_2$ concentration of the conversion reactions (R1-3). However, the reaction time has to be relatively short to avoid the interference of $HO_2$ recycling as mentioned by Berresheim et al., (2000). $E_{Ion}$ is affected by the flow dynamics, which determined by the mixing of flows, and the electric field inside the ionization region. The electric field forces the $NO_3^- \cdot (HNO_3)_m \cdot (HO_2)_n$ primary ions to the center of the region for $H_2SO_4$ ionization. $E_{Trans}$ is related to the $N_2$ buffer and induces an electric field in the pinhole area. On the other hand, $E_{Trans}$ is proportional to the transmission of the neutral molecule and particles from sampling air to the mass spectrometer system which deteriorates the measurement and damages the mass spectrometer. Thus, the optimization should take both transmission efficiency and protection function into consideration.

In this study, the CIMS was optimized before the field campaign. The detailed specification was shown in Table S2. To maximize the $E_{Conv}$, 5 sccm $SO_2$ was added from the front injectors to the sample flow and the [$SO_2$] was around 12 ppm in the sample flow. The $C_3F_6$ (2 sccm) was added from the rear injector to cease the reaction. The concentration of $C_3F_6$ in sample inlet was 1072 ppm. The sample flow was 3727 sccm and the sample flow rate was 55 cm/s which means the reaction time for OH conversion is around 47 ms.

The reaction time affects the positive bias of OH arising from $HO_2$ + NO in the inlet. To estimate this bias, Tanner et al., (1997) calculated the OH produced by the $HO_2$ recycling reaction under different NO conditions (from <60 ppt to 1-2 ppb) in the inlet by a box model. Their results showed that the positive bias of less than $0.5 \times 10^6$ cm$^{-3}$ with a 60 ms conversion time, and the bias does not increase with the increase of NO concentration. Thus, the conversion time of 47 ms in our study should further reduce such positive bias.

Compared to $C_3F_6$, the other scavenger gas, propane ($C_3H_8$), has a higher elimination rate. However, the purity of the $C_3F_8$ from most of the suppliers is not high enough for elimination, we chose $C_3F_6$ due to its stable quality. These parameters taking the $E_{Conv}$ and the $HO_2$ interference mentioned above into consideration.

Similarly, the sample/sheath flow ratio was adjusted to 0.3 and the voltages different between sample and sheath flow were adjusted to 48 V to achieve the maximum $E_{Ion}$. Finally, the buffer flow was 440 sccm and the Pinhole voltage difference was 30 V for a better $E_{Trans}$ and to prevent neutral molecules enter the mass spectrometer system as the same time.

Table S2. Technical details and specifications of the OH-CIMS

| Efficiency Related | Parameters | Gas | Values | Units | Specification for Measurement | Values | Units |
|---|---|---|---|---|---|---|---|
| $E_{Conv}$ | Front Injection | $SO_2$ (0.9%) | 5 | sccm | Sample Flow [$SO_2$] | 12 | ppm |
| | Pulse Valve | $N_2$ | 2 | sccm | Cycle Duration (OH) | 6 | mins |
| | | $C_3F_6$ (99.9%) | 2 | sccm | Elimination Rate (OH) | 92% | |
| | Rear Injection | $C_3F_6$ (99.9%) | 2 | sccm | Sample Flow [$C_3F_6$] | 1072 | ppm |
| | | $HNO_3$ | 10 | sccm | Reaction Time | 47 | ms |
| | Sample Flow | | 3.7 | slpm | Sample Flow Speed | 55 | cm/s |
| $E_{Ion}$ | Sheath Flow | Zero Air | 12.6 | slpm | Reynolds Number in Ionization Chamber | >4000 Turbulent flows | |
| | | $HNO_3$ | 10 | sccm | | | |
| | | $C_3F_6$ (99.9%) | 2 | sccm | Sheath Flow [$C_3F_6$] | 159 | ppm |
| | Total Flow | | 16.8 | slpm | Sheath Flow Speed | 25 | cm/s |
| | Sheath Voltages | | -80 | V | Voltages Difference for ionization | 48 | V |
| | Sample Voltages | | -32 | V | | | |
| $E_{Trans}$ | Buffer Gas | $N_2$ | 440 | sccm | Voltages Difference for transmission | 80 | V |
| | Buffer Voltages | | -70 | V | | | |
| | Pinhole Voltages | | -40 | V | | | |
| Cal | Calibration Flow | | 10 | slpm | Calibration Factor (Reagent ion: $NO_3^-$) | $1.21*10^{-8}$ OH·cm$^3$/Hz | |
| | Flow Speed | | 65 | cm/s | | | |
| | Product It Value | | $8.8*10^{10}$ photon/cm | | | | |
| Uncertainties | Sigma ($\sigma$) | | 2 | | Detection Limit | In lab | 1.5 |
| | Calibration | | 38% | | | Daytime | 10 |
| | Overall | | 44% | | | Nighttime | 7.7 |

Notes:

sccm – Standard cubic centimeters per minute.     slpm - Standard liter per minute.

ppm – Parts per million     V – Voltage

The zero air was produced by Model 111 Zero Air Supply (Thermo Fisher Scientific) with an air compressor.

Suppliers for $N_2$, $SO_2$, and $C_3F_6$: Scientific Gas Engineering Co. Ltd., HK.

The current manuscript provides sufficient references for the measurement principle of the instrument but is lacking explanations of how the specific calibration error, detection limits, and overall accuracy of the new instrument were determined. Which factors contributed how much to the total accuracy of 44 %? How was the detection limit calculated and why was it larger in the field than in the laboratory? How large was the background compared to the OH signal? Were interference tests performed and what was the result? Were any corrections made for chemical interferences in the inlet as discussed by Berresheim et al. (2000)? These questions should be answered and supporting material could be presented in the Supplement. I also suggest moving Table 3 to the Supplement and giving there some explanations of its contents.

Response: Following the referee's suggestions, the overall accuracy and detection limit was added to the manuscript as follows:

"The averaged overall uncertainty for this campaign is 44% for OH measurement with consideration of the calibration accuracy and the variations in m/z at 62 ($S_{64}$, 18%) and at 97 ($TS_{97} - BS_{97}$, 13%) during observation (SE4 and SE5). Detailed descriptions of CIMS optimization, calibration process, calculation of detection limit, and the specifications are in the Supplementary Information."

The contribution of each factor to the calibration accuracy was presented in the supplementary as shown above.

The discussion on interference in inlet was added in the supplementary as follows:

"$E_{Conv}$ is dependent on the reaction time and the $SO_2$ concentration of the conversion reactions (R1-3). However, the conversion time has to be relatively short to avoid the interference of $HO_2$ recycling as mentioned by Berresheim et al. (2000).

…

The sample flow was 3727 sccm and the sample flow rate was 55 cm/s which means the reaction time for OH conversion is around 47 ms. The reaction time affects the positive bias of OH arising from $HO_2$ + NO in the inlet. To estimate this bias, Tanner et al., (1997) calculated the OH produced by the $HO_2$ recycling reaction under different NO conditions (from < 60 ppt to 1-2 ppb) in the inlet by a box model. Their results showed that the positive bias of less than $0.5 \times 10^6$ cm$^{-3}$ with a 60 ms conversion time, and the bias does not increase with the increase of NO concentration. Thus, the conversion time of 47 ms in our study should further reduce such positive bias."

The explanations of specific calibration error and detection limits were added to the updated supplementary as shown in the response of item 1. Please note that the detailed calibration procedures were moved from the manuscript to the supplements for a better reading experience.

The previous manuscript provided the reason for the larger detection limit in the fields. To make it clearer, we have rewritten these parts in the new manuscript as shown below:

"Due to variations in the concentrations of $H_2SO_4$ and other interference gases in the ambient air, the background signal during ambient measurement has a larger variation compared to the lab condition, resulting in a higher detection limit in the ambient condition."

The table was moved to the supplement as a summary of the added contents and to provide an overall specification of our CIMS. The optimization principle was also added to support the parameters in Table 3 and the table was revised for better clarity.

In the revised manuscript, we briefly describe the measurement principle, and key instrument parameters to support the measurement validity. The supplementary provides more detailed description of principles and values to support the measurement.

2. Measured diurnal OH profiles

The measured diurnal OH profiles in Figure 7 show plausible variations during daytime as can be seen from the correlation with solar UV and the OH model simulation. However, the considerable nocturnal OH concentrations between 0.5 x 10^6 cm-3 and 1 x 10^6 cm-3 (Figure 5, 6) are an order of magnitude larger than the simulation shown in Figure 7. The unexpectedly high nighttime values are not commented on or discussed. They could be due to a systematic instrumental offset or indicate real atmospheric OH at night. This needs to be discussed. For example, is there an instrumental baseline problem that cannot be eliminated by the chemical modulation in the CIMS inlet? Mauldin et al. (2012) reported a non-OH source of sulfuric acid in a Boreal forest (probably not applicable here) and Berresheim et al. (2014) found evidence for an unknown oxidant in coastal air that converts SO2 to sulfuric acid in their CIMS inlet. Could these unknown oxidants play a role in the measurements reported here? What would happen if the unknown oxidant chemistry in the instrument inlet would be influenced by the OH scavenger? Have you tried a different scavenger other than C3F6? If there is a problem with the baseline, it could potentially affect the daytime OH measurements as well. If the nocturnal OH levels observed by CIMS indicated true OH levels, this would be of considerable atmospheric relevance. How do the values compare to previous observations of nighttime OH in PRD (Lu et al., Atmos. Chem. Phys., 14, 4979–4999, 2014; Tan et al., 2019)?

Response: The systematic instrumental offset can be ruled out in our study since the non-OH-caused $H_2SO_4$ will not contribute to the OH-relevant signals due to the subtraction of the background signal ($BS_{97}$) from the total signal ($TS_{97}$). The $HO_2$ recycling reaction in different [NO] cause little bias as discussed below. Thus, these nighttime signals are the indication of real atmospheric OH at night. We have added discussion on section 2.2 about the other interferences as follows:

"Apart from ambient OH, some interference gases, such as ambient $H_2SO_4$, Criegee intermediates, and artificial OH produced by the ion source, can also be converted into $HSO_4^-$ and contribute to the signal $S_{97}$. To mitigate such interference, the scavenger gas ($C_3F_6$) and $N_2$ were added to the sample flow through electrically operated valves (see the pulsed flow in Figure 2) that automatically switched injection positions every 3 min."

The reason for the abnormally low simulated nighttime OH was due to the approach we adopted to constrain the $NO_x$ in the model. Following recommendation by Wolfe et al. (2016), the total $NO_x$ was constrained with the measurement, and the $NO/NO_2$ ratio was calculated by the model. In this case, the model calculated nighttime NO was close to zero which led to the abnormally low nighttime OH. In the revised version, we decide to constrain the NO and $NO_2$ independently, which should be more appropriate as each is accurately measured. This approach yielded nighttime OH concentration with no significant discrepancies from the measurement's nighttime. We note that the new simulation result further increases the discrepancy in the daytime. We have modified relevant figures as:

[Figure]

"Figure 6. Comparison between observed (dots) and simulated (lines) OH in the four continental cases (top panel) and the four coastal cases (lower panel), also showing measurement uncertainty (error bars) and $J_{NO2}$ measurement (yellow shades). The time zone was local time (+8 UTC) for the x-axis.

[Figure]

Figure 7. Diurnal profiles of average concentrations of measured (dots) with standard deviation and simulated (RUNBase, line) OH concentration, important trace gases and the measured BVOCs, AVOCs, OVOCs reactivity (BVOCs$_{Reac}$(M), AVOCs$_{Reac}$(M) and OVOCs$_{Reac}$(M)) for selected cases in continental (green) and coastal (blue) air masses. The grey shaded area denotes night-time. The error bars and shaded error bars are the standard deviation of the averaged data. The time zone was local time (+8 UTC) for the x-axis.

[Figure]

Figure 8. Diurnal profiles of measured (dots) with measurement uncertainty (error bars) and simulated (RUNBase, line) OH on 10 October 2020, with other chemical species and the measured BVOCs, AVOCs, OVOCs reactivity (BVOCs$_{Reac}$(M), AVOCs$_{Reac}$(M) and OVOCs$_{Reac}$(M)). The air mass drifted from continental (red) in the morning to

coastal (orange) in the afternoon. The grey shaded area denotes night-time. The time zone was local time (+8 UTC) for the x-axis."

The discussion about simulation and observation results of nighttime OH was added in section 3.2 as below:

"The below discussions focus on the comparison of the daytime results since the simulated night-time OH concentration was mostly within the measurement uncertainties and the night-time observations for Oct 08, 23, 27 and Nov. 5 were incomplete as shown in Figure 6."

We did try propane as the alternative scavenger gas, and it provides a higher removal efficiency of OH for background measurements. However, the purity of propane is not guaranteed by all the suppliers. Even though all the propane cylinders we purchased claim to have 99.95 vol.% purity, some of them work well on OH removal, others introduce larger interference. By increasing the $C_3F_6$ flows, high efficiency of OH removal can also be achieved. Thus, we use $C_3F_6$ as the scavenger in our system. We have added the flowing clarification in the supplementary:

"Compared to $C_3F_6$, propane ($C_3F_8$) has a higher elimination rate. However, the purity of the $C_3F_8$ from most of the suppliers is not high enough for elimination of the artificial OH, we thus chose $C_3F_6$ due to its stable quality."

Following the suggestion, we have added the following text for a comparison with the nighttime OH measurements from other groups in the PRD region:

"The averaged night-time OH concentrations in this study was $5.1 \pm 1.8 \times 10^5$ (1σ) cm$^{-3}$ which was comparable to the previous night-time results (below $10 \times 10^5$ cm$^{-3}$) measured at the PRD region (in Heshan, Tan et al., 2019, and in PKUSZ sites, Yang et al., 2022). The OH concentration was slightly higher in the evening ($6.8 \pm 1.1 \times 10^5$ (1σ) cm$^{-3}$) than that ($3.7 \pm 0.7 \times 10^5$ (1σ) cm$^{-3}$) in the morning, which might be due to the higher ozone concentration in the evening which leads to a higher OH production from alkene ozonolysis."

3. Comparison of modeled and observed OH concentrations

The authors report agreement with the modeled and measured OH in continentally influenced air and find that the model overestimates the observed OH in the coastal air. What can be learned from this result? Unfortunately, measurements of HO2 concentrations and OH reactivity were not performed in this campaign. Measurements of these quantities have become standard in most field campaigns over the past decade and are absolutely essential if new insights into atmospheric OH chemistry are to be gained. For example, field studies have shown that the agreement between modeled and measured OH can be misleadingly good. Kanaya et al. (Atmos. Chem. Phys., 12, 2567–

2585, 2012) and Whalley et al. (2018) reported missing OH production in their MCM models that were coincidentally compensated for by the model's overprediction of HO2, resulting in good agreement between modeled and observed OH concentrations. These model deficiencies were only detected because HO2 and OH reactivity measurements were available as additional constraints. These two parameters are of paramount importance for the understanding of OH since they dominate the chemical OH budget in most cases. For the same reasons, it is not clear if missing OH reactivity is the major reason for the overestimated modeled OH in the coastal air. Without knowing how well the model reproduces HO2, it is difficult to quantify the amount of missing OH reactivity. The authors assume that unknown atmospheric trace gases react with OH and form products that do not undergo further reactions (page 13, lines 23-24). The assumption that the products are inert is not very likely. Missing OH reactivity is most probably caused by unmeasured VOCs or OVOCs, which produce RO2 and HO2 when they react with OH. The additional peroxy radicals recycle some OH and thereby increase its total production rate (called P_constrain in Eq 9). It means that the required amount of missing OH reactivity is probably higher than the authors' estimate which is based on a fixed OH production rate. Here, additional HO2 measurements are missing to determine the total OH production independent of model assumptions.

Response: We agree that it is difficult to quantify the amount of the missing OH reactivity as explained by the referee. Thus, instead of quantifying the missing reactivity, the revised article will focus more on the qualitative discussion. Additional comparison and discussion about the simulated $HO_2$ and its sources and sinks were added. The estimated missing OH reactivity is considered as the minimum value of the missing reactivity.

The discussion of the $HO_2$ was added in section 3.3.2 before the discussion of the $HO_2$ heterogeneous uptake as below:

"Our calculated OH budgets show that the main sources of OH in the coastal air masses were the $HO_2 + NO$ reaction (69%), $O_3$ photolysis (14%), HONO photolysis (7%), and the reaction between ozone and $HO_2$ (4%). In the simulation, NO, HONO, and $O_3$ were constrained by observations. Could $HO_2$ be overestimated which would cause overprediction of OH?

The main $HO_2$ sources are the VOCs oxidation by OH and the photolysis of OVOCs. In our study, VOCs and OVOCs were more likely under-measured than over-measured, which would underpredict $HO_2$ rather than overpredicting it. In addition, not including the halogen chemistry would under-simulate $HO_2$ at this site (Peng et al., 2022; Xia et al., 2022).

We next examine the possibility of the underestimation of $HO_2$ sinks as the cause of the overprediction of OH. The major sinks of $HO_2$ include the reaction of NO to recycle OH, self-reaction to form $H_2O_2$, and heterogeneous loss by aerosol uptake. The first and

second pathways have been considered in the MCM. The heterogeneous uptake of $HO_2$ onto aerosol ……"

We also agree that the assumption for inert products is not very likely. See additional notes added after the $k_{miss}$ calcualtion as below:

"The calculated $k_{miss}$ could be a lower limit of the possible missing source as the products from the reactions of OH with unknown species are most likely to further react with the missing source to produce $RO_2$ and $HO_2$ and recycle back to OH."

4. Quantification of VOCs

The total amount of VOCs is expressed in many places in the paper as the sum of the VOC mixing ratios (ppb). While the total mixing ratio is a useful quantity to indicate the amount of measured organic carbon, it tells us little about its relevance for OH chemistry. Since the rate constants for different VOC species may differ by order of magnitude, it is better to report the total organic OH reactivities of the measured VOCs and their subgroups (AVOC, BVOC, etc.) to characterize the chemical conditions.

Response: We agree with these comments and have now used the reactivity of the measured VOCs and their subgroups (AVOCs, BVOCs, and OVOCs) to characterize the chemical condition throughout the manuscript. The changed figures including the Figure 7, 8 presented above and the Figures 3 and 4, as shown below.

[Figure]

Figure 3. Time series of OH between 7 October and 23 November with measured weather conditions (temperature and RH), OH primary sources (ozone and HONO), $NO_x$ (NO and $NO_2$), reactivity of measured VOCs and OVOCs ($VOCs_{Reac}(M)$ and $OVOCs_{Reac}(M)$), and photolysis frequency ($J_{NO2}$). All measurement data shown are 10 min averages. The gaps of the data were due to the calibration or instrument maintenance. The black lines separate the non-continuous days during measurement. The grey shaded area denotes night-time.

[Figure]

Figure 4. Diurnal profiles of the average (±1σ) concentrations of OH, other chemical species, the measured VOCs reactivity and OVOCs reactivity ($VOCs_{Reac}(M)$ and $OVOCs_{Reac}(M)$), and meteorological parameters ($T$, RH, $J_{NO2}$) during the field campaign. The grey shaded area denotes night-time. The error bars and shaded error bars are the standard deviation of the averaged data.

**5. Ozone photolysis frequency**

The photolysis of ozone-forming O(1D) is one of the major processes that produce HOx. The corresponding j-value is calculated in the present work by using a clear-sky parametrization from Saunders et al. (2003). The values are then scaled with the ratio of modeled-to-measured jNO2 (to correct for cloud effects?). The whole approach has a considerable error that is not discussed in the paper. jO1D depends on the total atmospheric ozone column and air temperature, which are both not considered in the parametrization. The parameterization is useful for pure modeling studies, but not a good choice for the description of real ozone photolysis frequencies in a field campaign. Ideally, jO1D is measured as is done by many groups. The next best approach would be to simulate the clear-sky values by a radiative transfer model (for example by the freely available Tropospheric Ultraviolet and Visible (TUV) Radiation Model from NCAR) taking total ozone and temperature into account. Also, note that jO1D responds differently to cloudiness compared to jNO2 (see for example, Walker et al., Environ. Sci.: Atmos., DOI: 10.1039/d2ea00072e). The authors should attempt to estimate a more realistic jO1D by means of the TUV or a similar radiative transfer model.

Response: Thanks for the comments. We agree that we should use the TUV model and consider the cloudiness effect to estimate a more realistic JO1D.

In the updated version, we used the TUV model (v5.2) in F0AM for clear-sky j-value simulation. The Ozone column we used for TUV calculation is 240 DU which is the average number from October to November 2020 for the Hok Tsui area according to the worldview website (EOSDIS Worldview (nasa.gov)).

The simulated JO1D result will still be corrected by measured $jNO_2$ for the following reasons.

a) The correction factors for different j-values presented by recent research (Walker et al., 2022) were measured in the UK, and the factors were the average values of the period from December to March. This means the factor is not suitable for our study (different latitudes, and different seasons).
b) Secondly, the coastal cases (Oct-25 to 27 and Nov05) that show discrepancy are mostly sunny days, which means the cloudiness effect on jO1D in this study is not as significant as discussed in Walker's article.

We have added the following text in section 2.3:

"The photolysis frequencies for other species were calculated by the "HYBRID" method in F0AM which is based on Tropospheric Ultraviolet and Visible (TUV v5.2) Radiation Model from National Center for Atmospheric Research. The Ozone column we used for TUV calculation was 240 DU (the Dobson unit) which is the average number from October to November 2020 for the Hok Tsui area according to the worldview website (EOSDIS Worldview (nasa.gov)). The simulated photolysis frequencies were then scaled by the correction factor obtained from the comparison between observed and modeled $J_{NO2}$."

And add further discussion in the section 3.3.1

"On the first possibility, we acknowledge that the correction factor for photolysis frequencies due to cloud presence may be different for different species (Walker et al., 2022), thus, using a single correction factor (based on $J_{NO2}$) may introduce uncertainty in the model simulations. We think such uncertainty should not be significant because the weather was mostly sunny in the coastal cases."

6. Literature review

In the introduction of the paper, the authors present Table 1 for an overview of previously published comparisons between modeled and measured atmospheric OH. The table takes up a large part of the paper but is not very informative due to the lack of its discussion. In order to judge the listed comparison results, detailed explanations would be needed on how the past measurement techniques and chemical models have improved over the last 2-3 decades. To keep the paper focused, I suggest removing Table 1 and Figure 1. It is sufficient to refer to corresponding review articles (e.g.,

Heard and Pilling, Chem. Rev. 2003, 103, 5163-5198; Stone et al., 2012; Rohrer et al., 2014; Lu et al., National Science Review 6: 579–594, 2019).

Response: Thanks for the comments, we agree that Figure 1 and Table 1 in the main text may distract the readers. But we still think the summary of the previous studies is useful to the reader for comparing the concentration and modelling result from the previous studies. The other referee had positive comment on Figure 1 and Table 1. We decided to move them to the supplementary.

Specific comments

1. Abstract and Introduction. OH, reactivity and missing OH reactivity need to be defined.

   Response: The definition was added as:

   "The missing OH reactivity, which is defined as the OH loss from unmeasured trace gases, is proposed to be the cause of this overprediction."

2. Page 2 – line 4. The hydroxyl radical (OH) dominates ...

   Response: Suggestion adopted.

3. Page 2 – line 9. The sentence can be deleted since heterogeneous losses of OH on particles are generally excluded from atmospheric models. If you want to keep the sentence, you may explain that OH is too short-lived to experience significant loss by collisions with particles.

   Response: The sentence was deleted.

4. Page 2 – line 22. The paper by Hard et al. 1979 is not a good choice to inform interested readers about LIF techniques that have been used internationally in the last three decades. Hard et al. were pioneers of the FAGE concept, but the laser spectroscopy described in their 1979 paper used 282 nm excitation that was significantly affected by self-generation of OH and abandoned later (see an overview by Crosley, J. Atmos. Sci. 1995, Vol 52 No 19, 3299). More recent descriptions can be found, for example, in Heard and Pilling, Chem. Rev. 2003, 103, 5163-5198 and Schlosser et al., Atmos. Chem. Phys., 9, 7923–7948, 2009.

   Response: Thanks for the suggestion, the reference Hard et al., 1979 was replaced by Heard and Pilling et al., 2003.

5. Page 3 – lines 5-6. OH, underprediction is not restricted to biogenic environments but was also observed in urban atmospheres when NO was below 1 ppb. See, for example, Whalley et al., 2018 and Tan et al., 2017, 2019.

Response: The sentence was modified as below:

"Few recent studies also found the OH estimation in urban atmospheres when the NO level was below 1ppb (Whalley et al., 2018; Tan et al., 2017, 2019)."

6. Page 3 – line 12. The results in the Zhang et al. (2006) paper are outdated because the OH data used had a significant calibration error that was later corrected. The revised TRACE-P data show good agreement with model results (Ren et al., Journal Geophysical Research, Vol. 113, D05310, doi:10.1029/2007JD009166, 2008).

Response: Thanks for the comments, we have removed this reference since it shows good agreement with simulation and not suitable for the content.

7. Page 3 – lines 8-23. Specify the locations or types of environments for which OH was overestimated by models.

Response: Suggestion accepted, and the contents changed as below:

"Unmeasured VOCs have been proposed as the missing OH sinks, resulting in the overestimation of OH, e.g., in aged air in Idaho Hill (McKeen et al., 1997), in the marine boundary layer (MBL) of Mace Head (Carslaw et al., 1999; Berresheim et al., 2002), in the MBL in Tasmania (Creasey et al., 2003), in Antarctica (Mauldin III et al., 2010), and the urban area of California (Griffith et al., 2016). Other studies have shown evidence of missing OH sinks in various types of environments (Lou et al., 2010; Yang et al., 2016), e.g., in the clean forest (Hansen et al., 2014) and marine (Thames et al., 2020) areas, which is likely resulted from unmeasured organic compounds in emitted from biogenic (Kaiser et al., 2016) or oceanic (Thames et al., 2020) sources and their oxidation products."

This information was also presented in previous table 1 and more informative in previous figure 1 as the types of environments were labelled in different colors. The Figure 1 and Table 1 were moved to supplementary in the updated version, as explained in the response to general comment #6.

8. Page 3 – lines 16-19. It should be mentioned that numerous studies in all types of environments (marine, rural, forest, urban) have demonstrated missing OH reactivity by direct measurements of OH reactivity that were compared to the reactivity calculated from measured VOCs. Overviews can be found, for example, in Lou et al. Atmos. Chem. Phys., 10, 11243–11260, 2010, and Yang et al., Atmospheric Environment 134 (2016) 147-161.

Response: The description of the OH reactivity studies was changed as follows:

"Other studies have shown evidence of missing OH sinks in various types of environments (Lou et al., 2010; Yang et al., 2016), e.g., in the clean forest (Hansen

et al., 2014) and marine (Thames et al., 2020) areas, which is likely resulted from unmeasured organic compounds emitted from biogenic (Kaiser et al., 2016) or oceanic (Thames et al., 2020) sources and their oxidation products."

9. Page 3 – lines 19-23. Which measurement techniques are discussed here?

Response: The techniques were added to the contents as shown below:

"A few studies have shown that the overestimations fall within measurement uncertainties of DOAS, CIMS (McKeen et al., 1997), and LIF, (Carslaw et al., 1999) while others have suggested a possible sampling loss of OH in CIMS (Mauldin III et al., 2010) or a possible calibration bias due to low relative humidity (CIMS, Mauldin III et al., 2001)."

10. Page 7, line 12. The value for the absorption cross-section of H2O for the Hg 184.9 nm line given by Cantrell et al. is 7.14 x 10-20 cm (not 7.22 x 10-20 cm2).

Response: Thanks for the careful check. The number was revised and moved to the supplementary due to the change of structure mentioned in item 1.

11. Page 8 – lines 4-6. The measurement site was located on a coast, where halogen chemistry could play a role in OH. Since it is not mentioned, I assume that halogen chemistry was not included in the model? How much could your model results be affected by BrO or IO if you assume mixing ratios as reported in the literature (e.g., Fan et al., Atmos. Chem. Phys., 22, 7331–7351, 2022).

Response: The halogen chemistry was not included in the model. The inclusion of halogen do not affecting the discrepancy for OH significantly. We added the discussion of halogen in the section 3.3.1 when discussing the uncertainties of simulation.

"We did not include halogen chemistry in our study as we wanted to compare our results with previous modelling work, most of which did not consider halogen chemistry. Our other studies at the same site that did consider the halogen chemistry show a 4% increase in OH concentration from Cl chemistry (Peng et al., 2022) and 2.8% from Br chemistry (Xia et al., 2022), which would even increase the model-measurement discrepancy in the coastal air mass."

12. Page 8 – line 7. Which HOx recycling mechanisms do you mean?

Response: The mechanisms that generate OH by $HO_2$. The sentence was revised as below:

"The isoprene degradation mechanisms, and in particular the mechanisms of OH regenerated by $HO_x$ recycling in low $NO_x$ condition, were improved in MCM v3.3.1."

13. Page 9 – line 4. Molecular weight must be molecular mass.

    Response: Revised.

14. Page 9 – lines 8-10. I do not understand how the authors constrained their model with NO and NO2. Did you use measured values or concentrations that were calculated from a PSS assumption? As NO and NO2 have a strong influence on OH, the handling of the NOx data should be explained in detail. Figure S1 needs to be explained and its relevance should be discussed.

    Response: As mentioned in item 2, we re-run the model by constraining the NO and $NO_2$ independently. The previous Figure S1 was removed as it is no longer relevant.

15. Page 9 – lines 11-12. How sensitive are the modeled OH concentrations to the assumed VOC concentrations?

    Response: The sensitivity tests figures were added in the supplementary figure S4.

[Figure]

Figure S4. Sensitivity tests for the simulated OH and $HO_2$ in continental and coastal cases and on 10 October. $RUN\gamma_{MAX}$ shows the simulated results for the maximum heterogeneous uptake effect of $HO_2$ ($\gamma = 1$). The $RUN_{VOC0}$ and $RUN_{VOCDL}$ show the simulated results that constraints "0" and the detection limit value as the concentration of VOCs when their concentration were below detection limits.

The discussion about the assumed VOCs concentration was added in Section 3.3.1 as follow:

"Regarding the uncertainty from the VOCs input, we conducted a sensitivity test to show that the treatment of VOCs that were below the detection limits should have a negligible effect on OH simulation ($RUN_{VOC0}$ and $RUN_{VOCDL}$ in Figure S4)."

16. Page 9 – lines 17-19. In Table 4, the OH production and destruction are calculated from concentrations of species that were not measured (e.g., H2, H2O2). How were the concentrations of these species determined? Did the model consider the deposition of products that are built up in the model over three days? Was the three-day spin-up simulated with constant photolysis frequencies and constant deposition?

Response: The $H_2$ concentration was set to 550 ppb as recommended by Wolfe et al., 2016. The $H_2O_2$ is calculated from the three-day spin-up and the concentration was stabilized during the spin-up.

We added notes in table 2 as below:

"Notes: The $H_2$ concentration were constrained as 550 ppb in the model simulation.
The $H_2O_2$ were simulated by model with averaged concentration at 0.95 ppb."

The spin-up was simulated with constant photolysis and deposition. The explanation of the spin-up was added in the revised manuscript, as follows:

"For each run, a three-day spin-up was performed with constant photolysis and deposition to create a stable model environment and to avoid the uncertainty of unconstrained species (Carslaw et al., 1999)."

17. Page 10 – lines 1-2. In the literature, it is more custom to look at the correlation between jO1D and OH (e.g., Ehhalt and Rohrer, J. Geophys. Res. 105, No. 03, 3565-3571, 2000; Rohrer and Berresheim, 2006; Ma et al., 2019; Tan et al., 2017). How comparable is your correlation for jO1D?

Response: The comparison of jO1D and OH was added in the revised manuscript. The jO1D shows a similar correlation with OH compared to JNO2.

The contents in section 3.1 were changed as below:

"The OH concentrations showed a distinct diurnal pattern and a positive correlation with $J_{NO2}$ and calculated $J_{O1D}$ ($R^2 = 0.68$ and 0.46 for $J_{NO2}$ and calculated $J_{O1D}$, respectively, Figure S2)."

Figure S2 was revised as below:

[Figure]

Figure S2. Correlation between observed OH concentration and a) photolysis frequency of $NO_2$ ($J_{NO2}$), and b) model simulated photolysis frequency $O_3$ (Simulated $J_{O1D}$). The linear regressions with respect to total, coastal, and continental cases are labelled in black, blue, and green. Note that the coastal and continental cases are reported as correlations for all cases in different clusters, not only the selected cases in the Figure 8 comparison.

18. Page 10 – line 29. What is a continental air mass cluster?

   Response: It should be 'continental air mass'

19. Page 11 – lines 11-26. It would be of interest to see also plots of the model results for HO2 and the OH reactivity, which both have a large influence on OH. How does the simulated OH reactivity compare to the reactivity that can be calculated from

measurements (CO, NOx, VOCs, OVOCs)? How does the modeled HO2 concentration compare to observations in PRD (e.g., Tan et al., 2019)?

Response: The comparison of simulated $HO_2$ and OH reactivity with other studies was added in the manuscript.

The simulated $HO_2$ concentration was added to different results sections as shown in the response to general comment # 3.

The difference between measured and model simulated OH reactivity was shown in Figure S5 as below:

[Figure]

"Figure S5. Simulated reactivity for continental cases (a), coastal cases (b), and 10 October (c). The AVOCs, BVOCs, OVOCs, and Inorganic demostrates the reactivity calculated from the measured species and the Model Calculated represstens the reactivity calculated by the derived species simulated by model."

20. Page 12 – lines 3-4. The meaning of '$p < 0.05$' needs to be explained.

Response: It means these two data sets are statistically significant difference. The explanation for '$p < 0.05$' was added in the section 3.2.2.

"The coastal air masses showed statistically significant ($p < 0.05$) lower $NO_x$ (−63%), AVOCs (−47%), BVOCs (−50%), OVOCs (−23%), and CO (−31%) concentrations compared with the continental cases (Figure 8, Table S1)."

21. Page 12 – line 24. Table 4 does not fit here. Do you mean Table S1?

Response: Sorry for the mistake, it should be Table S1 in previous manuscript and the Table S3 in the revised manuscript.

22. Page 18 – line 28. Check the author list.

Response: Sorry for the mistake, the author list as changed as follow:

"Eisele, F. L. and Tanner, D. J.: Ion-assisted tropospheric OH measurements, J. Geophys. Res., 96, 9295, https://doi.org/10.1029/91JD00198, 1991."

23. Figure 1: I suggest removing this figure and Table 1 as explained above.

Response: We would like to keep them but move them to the supplementary as explained above.

24. Figure 4. Use different colors to make it easier to distinguish the NO and NO2 data. It is difficult to see any temporal structure in the shown NO2 photolysis frequencies. I suggest drawing jNO2 as a solid line. CO, HCHO, jO1D, and the total reactivity of VOCs and OVOCs should be shown since they have an important influence on OH. What is the reason for the gaps in the OH time series? The OH concentrations are determined from the difference between two noisy signals (Eq. E1). Near the limit of detection, the difference can become occasionally negative. Were any negative OH data calculated that are not shown in the figure?

Response: The color and the line were changed as suggested. The VOCs and OVOCs reactivities were used in the revised as shown above. The reason for gaps in the OH time series was due to the calibration and the instrument maintenance. The explanation was added in Figure caption:

"The gaps of the data were due to the calibration or instrument maintenance."

The negative values in the updated manuscript were shown by setting the range of the y-axis from 0 to $-1 * 10^6$. The revised figure 3, and figure 6 were shown above in general comment # 2.

25. Figure 5. As mentioned before, it would be good to see jO1D and the total OH reactivities of VOCs and OVOCs. How large were the NO nighttime values?

Response: The total OVOCs and VOCs concentrations have been changed to their OH reactivities as the Figures shown above. The JO1D on the other hand is not suitable for Figure 4 (Figure 5 in previous version) because this is the figure shown measurement results and the JO1D were simulated by model. The figure S2 shows the correlation of simulated JO1D and OH observation as discussed above.

The average nighttime NO values were 0.146 ppb.

26. Figure 7 + 8. Explain the shown error bars of measured OH. Do they denote precision (signal noise) or errors of calibration? Here and in other figures: which time zone is used for the x-axis? It would be more informative to see the OH reactivity of AVOC, BVOC, and OVOC plotted rather than the total sum of mixing ratios.

Response: Thanks for the comments. The error bars were the measurement uncertainties in Figure 6 and the variation (standard deviation) of the OH measurement results within each measurement cycle (6 mins) in Figure 7. The time zone is local time (+8 UTC). These and the VOC reactivities have been added.

Please find the figures caption changes in the response of general comments 2.

27. Table 2. Some of the measured trace gases that are used as model input (Table S1) are missing and should also be listed (e.g., CO, SO2, NH3). For which signal-to-noise ratio is the detection limit defined? The table should also specify the accuracies of the measurements.

Response: The Table 1 (Table 2 in previous manuscript) was updated as recommended.

Table 1. Measuring instruments and measured species in the field campaign

| Species | Instruments | Time Resolution | Detection Limit | Accuracy (1σ) |
|---|---|---|---|---|
| NO, NO$_2$ | Chemiluminescence/photolytic converter (Thermo, Model 42i) | 1 min | 60 ppt | NO: $\pm$ 5.2% NO$_2$: $\pm$ 15.2% |
| OH | Nitrate-quadrupole chemical ionization mass spectrometer (CIMS) | 10 s | Lab: $1.5 \times 10^5$ cm$^{-3}$ Daytime: $1 \times 10^6$ cm$^{-3}$ | $\pm$ 44% |
| Ozone | Ozone analyzer, model 49i, Thermo Scientific | 1 min | 0.5 ppb | $\pm$ 6.0% |
| JNO$_2$ | Filter Radiometer, Metcon | 1 min | $4 \times 10^{-5}$ s$^{-1}$ | $\pm$ 10% |
| HONO | Iodide-Tof-CIMS, Aerodyne Inc | 1s | 0.2 ppt | $\pm$ 15 % |
| SO$_2$ | Pulsed Fluorescence SO$_2$ Analyzer (Thermo, Model 43i) | 1 min | 1 ppb | $\pm$ 6.1% |
| CO | Gas Filter Correlation CO Analyzer (Thermo, Model 48i) | 1 min | 40 ppb | $\pm$ 7.4% |
| NH$_3$ | Chemiluminescence NH$_3$ Analyzer (Thermo, Model 17i) | 2 mins | 1 ppb | $\pm$ 8% |
| Particle number size distribution | Scanning mobility particle sizer, TSI | 5 mins | 1 particle cm$^{-3}$ | $\pm$ 10% |
| VOCs | GC-MS/FID (GC955 Series 611/811, Syntech Spectras) | 1 hour | ~10 ppt | $\pm$ 20% |
| | PTR-MS (PTR-QMS 500, IONICON Analytik, Austria) | 5 mins | 20 ppt | $\pm$ 20% |

28. Table 3. The table needs explanations. The listed quantities (e.g., sample flow [SO2], elimination rate, switching time, reaction time, etc.) should be explained and defined. What is the purpose of sample flow [NO]? What is the purpose of C3F6 in the sheath flow? What are the main gas components in the SO2 and NO mixtures? Specify the purity of the main components in the gas mixtures and of the other used gases (N2, zero air, HNO3). Who is the supplier of the gases? Define 'sccm'. Is the OH accuracy (44 %) given for 1 or 2 sigmas? Why is the value larger than in the text (38 %)?

Response: Thanks for the comments, the Table 3 in previous manuscript were moved to the supplementary and shown in previous response (general comment # 1)

The listed quantities like sample flow, [$SO_2$], [$C_3F_6$], and switching time were explained in the sensitivity optimization section in the updated supplementary.

The [NO] was added by mistake and was deleted in the updated version.

The OH measurement accuracy is 44% with 2 sigma which considers both calibration uncertainty and the measurement signal variation. The modification of the section 2.2 and the details on calculation and propagation added in supplementary were shown in the response to general comments # 1.

29. Table 4. O1D should read O1D+H2O. Which reaction is meant by CH3CCH3OOB and CH3CCH3OOC? A large fraction of 36-40 % of the OH loss is attributed to the 'Other' species. Which species are lumped in 'Other'?

Response: The O1D were changed to O1D+H2O as recommended.

The CH3CCH3OOB was the excited form of CH3CCH3OOC based on the MCM official website (MCM Website CH3CCH3OOB (leeds.ac.uk) and MCM Website CH3CCH3OOC (leeds.ac.uk)) They came from different reactions even though they share the same structure. CH3CCH3OOB was produced from the ozonolysis of 2-Methyl-2-butene and CH3CCH3OOC was produced from the ozonolysis of 2,3-Dimethyl-2-butene. In the updated table 2 and Figure S3, for better understanding, we changed these two species to DM23BU2ENE + $O_3$ and ME2BUT2ENE + $O_3$, respectively.

We plot pie charts to explain the species included in "Other" items as shown below:

[Figure]

Since the "Other" items include a large number of species that contribute less than 2% to the OH reactivity each, it would not be informative to show them. Thus, the pie charts were not included in the revised manuscript. Instead, the explanation for "Other" was added in the note of Table 2. The number in Table 2 and Figure S3 have also been changed due to the rerun of the model.

"Table 2. OH budgets for the selected continental and coastal cases, morning and afternoon of 10 October.

| Continental Case | | Coastal Case | | Oct 10 Morning | | Oct 10 Afternoon | |
|---|---|---|---|---|---|---|---|
| Production | | | | | | | |
| $HO_2$ + NO | 77.66% | $HO_2$ + NO | 69.02% | $HO_2$ + NO | 73.17% | $HO_2$ + NO | 65.11% |
| $O^1D$ +$H_2O$ | 7.98% | $O^1D$ +$H_2O$ | 13.80% | $O^1D$ +$H_2O$ | 10.73% | $O^1D$ +$H_2O$ | 15.04% |
| HONO + hv | 5.78% | HONO + hv | 7.32% | HONO + hv | 8.65% | HONO + hv | 7.16% |
| $HO_2$ + $O_3$ | 1.97% | $HO_2$ + $O_3$ | 3.60% | $HO_2$ + $O_3$ | 1.70% | $HO_2$ + $O_3$ | 3.80% |
| DM23BU2ENE +$O_3$ | 1.59% | ME2BUT2ENE + $O_3$ | 1.40% | $H_2O_2$ + hv | 0.52% | $H_2O_2$ + hv | 1.63% |
| Other | 5.02% | Other | 4.85% | Other | 5.23% | Other | 7.25% |
| Loss | | | | | | | |
| CO | 19.91% | CO | 23.39% | $C_5H_8$ | 15.96% | $C_5H_8$ | 15.38% |

| | | | | | | | |
|---|---|---|---|---|---|---|---|
| $NO_2$ | 9.38% | $C_5H_8$ | 8.17% | CO | 14.68% | CO | 13.72% |
| $C_5H_8$ | 9.09% | $C_2H_5CHO$ | 7.44% | $CH_3CHO$ | 8.76% | $C_2H_5CHO$ | 10.64% |
| $C_2H_5CHO$ | 7.96% | $CH_3CHO$ | 6.97% | $C_2H_5CHO$ | 8.31% | $CH_3CHO$ | 7.52% |
| $CH_3CHO$ | 7.94% | $NO_2$ | 6.27% | $NO_2$ | 5.70% | HCHO | 3.44% |
| $CH_4$ | 3.68% | $CH_4$ | 5.91% | $CH_4$ | 3.04% | $NO_2$ | 3.33% |
| HCHO | 2.79% | HCHO | 2.50% | HCHO | 3.03% | $CH_4$ | 3.06% |
| ACR | 1.41% | $O_3$ | 2.04% | ACR | 1.65% | ACR | 1.71% |
| $HOCH2CHO$ | 1.36% | $H_2$ | 1.71% | $HOCH_2CHO$ | 1.61% | $HOCH_2CHO$ | 1.71% |
| Other | 36.48% | Other | 35.61% | Other | 37.27% | Other | 39.50% |

Notes: The $H_2$ were constrained as 550 ppb in the model simulation.

The $H_2O_2$ were simulated by model with averaged concentration at 0.95 ppb.

ACR- acrolein                    HCHO: Formaldehyde

$C_5H_8$: Isoprene                 $HOCH_2CHO$: Glycolaldehyde

$C_2H_5CHO$: Propanol              $CH_3CHO$: Acetaldehyde

ME2BUT2ENE: 2-Methyl-2-butene     DM23BU2ENE: 2,3-Dimethyl-2-butene

Other represents the group of the species contribute less than 2% to the total OH reactivities. Most of them were the intermediate species produced by the reaction of OH with VOCs.

[Figure]

Figure S3. OH radical budgets for the continental cases, coastal cases, and 10 October. Where DM23BU2ENE and ME2BUT2ENE represent 2,3-Dimethyl-2-butene and 2-Methyl-2-butene respectively."

30. Figure S3 + S4. Remove the labels 'good matched' and 'overestimated'. Which species are included in 'Other'? The OH reactivity shown in Figure S4 is better called simulated reactivity or modeled reactivity. Then it is clear that the reactivity includes the contributions of measured and modeled species as well.

Response: The labels in Figures S3 and S5 (Figures S3 and S4 in previous manuscript) were revised as recommended. The 'Other' item was changed to 'Model Calculated' to show the reactivity that included in simulation but not observed in the campaign in Figure S5 as shown in previous response for item 20. In this way, the figure shows the reactivity of measured species (AVOCs, BVOCs, OVOCs and Inorganic) and the reactivity of model calculated species.

31. Table S1 is difficult to understand. What are the units of the concentration measurements? All symbols and abbreviations (e.g., OH_DL, PM_SUR) need to be defined. Unusual names for organic compounds, e.g. IC4H10, EBENZ, TEXs_PTR, C5H8deri, should be replaced by their chemical names. Indicate for each organic species which technique (GC or PTR-MS) was used for its measurement. Indicate which of the listed parameters/species were actually used as model input.

Response: The names of the abbreviations have been defined in the new version of table S3. Labels have been added to identify the species that were constrained in the model. The revised table with notes shown as below:

Table S3. Average concentration with standard deviation of measured species with respect to different cases

| Species Abb. | Species Name | Total | Coastal | Continental | Oct10M | Oct10A | Episode |
|---|---|---|---|---|---|---|---|
| OH $10^6$ (cm$^{-3}$) | Hydroxyl radical | 2.4±1.9 | 2.5±1.4 | 3.1±1.7 | 3.7±2.1 | 1.8±1.5 | 4.2±2.8 |
| OH_DL $10^6$ (cm$^{-3}$) | Detection limit of hydroxyl radical | 1.0±0.5 | 0.8±0.3 | 0.9±0.3 | 1.2±0.5 | 1.5±0.7 | 1.0±0.5 |
| OH_Err $10^6$ (cm$^{-3}$) | OH Measurement Uncertainty | 1.5±1.0 | 1.8±0.5 | 1.7±0.6 | 0.9±0.8 | 1.0±0.9 | 2.5±1.7 |
| PM_Num $10^3$ (#/cm$^3$) | Number of particulate matters | 3.8±1.9 | 4.1±1.7 | 4.9±1.4 | NaN | NaN | 5.6±2.0 |
| *PM_Sur $10^7$ (nm$^2$/cm$^3$) | Surface of particulate matters | 19.7±9.0 | 15.0±2.3 | 26.8±4.3 | NaN | NaN | 31.5±14.2 |
| PM_Vol $10^9$ (nm$^3$/cm$^3$) | Volumn of particulate matters | 7.6±3.8 | 4.9±0.7 | 10.5±1.5 | NaN | NaN | 12.0±5.9 |

| | | | | | | | |
|---|---|---|---|---|---|---|---|
| *RH (%) | Relativie humidity | 70.1±10.1 | 69.9±4.5 | 64.2±2.8 | 69.3±4.6 | 63.7±3.7 | 61.6±9.6 |
| *CO2 (ppm) | Carbon dioxide | 426.7±14.8 | 412.8±1.2 | 426.3±2.4 | 424.1±2.8 | 425.2±2.5 | 428.0±10.8 |
| WindDi (°) | Wind direction | 45.9±35.7 | 49.3±0.9 | 53.3±24.0 | 30.7±5.5 | 48.5±3.3 | 125.7±90.1 |
| WindSp (m/s) | Wind speed | 4.3±1.6 | 5.2±0.9 | 3.9±0.6 | 4.0±0.5 | 3.0±0.5 | 2.4±1.5 |
| *Temp (°C) | Temperatuer | 23.3±3.5 | 24.7±0.9 | 25.5±1.4 | 25.3±1.6 | 27.4±0.9 | 26.7±2.1 |
| [†]SO2 | Sulfur dioxide | 2.6±1.2 | 3.2±0.2 | 3.4±0.1 | 3.5±0.2 | 3.2±0.1 | 4.4±0.8 |
| [†]CO | Carbon monoxide | 304.9±72 | 217.4±10.9 | 318.0±8.5 | 291.3±16.3 | 258.4±14.1 | 329.0±74.6 |
| NH3 | Ammonia | 8.8±1.8 | 8.9±0.4 | 9.5±0.6 | 9.7±0.2 | 9.2±0.6 | 10.6±3.0 |
| [†]NO | Nitrogen Monoxide | 0.9±1.4 | 0.3±0.1 | 0.7±0.4 | 0.6±0.3 | 0.3±0.1 | 1.4±1.3 |
| [†]NO2 | Nitrogen Dioxide | 3.9±3.5 | 1.6±0.7 | 4.5±1.1 | 3.4±1.4 | 1.1±0.5 | 10.1±5.6 |
| [†]NOx | Nitrogen Oxides | 4.8±4.4 | 1.9±0.7 | 5.2±1.2 | 4.0±1.6 | 1.4±0.5 | 11.4±6.2 |
| [†]O3 | Ozone | 49.9±20.6 | 59.5±10.1 | 54.7±14.5 | 44.2±9.9 | 61.2±3.8 | 70.4±33.5 |
| [†]JNO2 10[-3] (s[-1]) | Photolysis rate constant of NO2 | 3.6±2.5 | 4.7±2.4 | 4.0±2.0 | 4.8±2.5 | 5.0±2.6 | 4.3±2.2 |
| ^HONO | Nitrous acid | 0.15±0.069 | 0.15±0.019 | 0.16±0.035 | 0.29±0.101 | 0.14±0.015 | NaN |
| *C2H4 | Ethene | 1.4±1.3 | 0.5±0.1 | 0.7±0.1 | 0.6±0.1 | 0.3±0.1 | 0.9±0.2 |
| *C2H6 | Ethane | 1.9±0.9 | 1.4±0.1 | 2.1±0.1 | 2.0±0.1 | 1.7±0.1 | 2.3±0.5 |
| *C3H8 | Propane | 1.7±0.9 | 1.1±0.2 | 1.5±0.2 | 1.3±0.1 | 0.8±0.1 | 2.1±1.7 |
| *C3H6 | Propene | 0.10±0.05 | 0.07±0.01 | 0.11±0.02 | 0.18±0.06 | 0.06±0.01 | 0.12±0.04 |
| *C2H2 | Ethyne | 1.63±0.65 | 0.97±0.03 | 1.42±0.23 | 1.07±0.08 | NaN | 1.39±0.48 |
| *IC4H10 | i-Butane | 0.55±0.44 | 0.22±0.04 | 0.61±0.14 | 0.44±0.09 | 0.23±0.07 | 1.02±1.04 |
| *NC4H10 | n-Butane | 0.76±0.60 | 0.27±0.06 | 0.88±0.19 | 0.67±0.13 | 0.32±0.08 | 1.53±1.62 |
| *TBUT2ENE | But-2-ene | 0.06±0.01 | 0.05±0.00 | 0.06±0.00 | 0.05±0.00 | NaN | 0.06±0.01 |
| *BUT1ENE | But-1-ene | 0.08±0.03 | NaN | 0.10±0.01 | 0.08±0.01 | NaN | NaN |
| *IC5H12 | i-Pentane | 0.40±0.22 | 0.18±0.04 | 0.42±0.05 | 0.46±0.03 | 0.28±0.11 | 0.60±0.36 |
| *NC5H12 | n-Pentane | 0.24±0.12 | 0.13±0.02 | 0.24±0.02 | 0.33±0.05 | 0.17±0.04 | 0.29±0.21 |
| *C4H6 | Buta-1,3-diene | 0.06±0.01 | NaN | 0.06±0.00 | NaN | NaN | 0.06±0.00 |
| *M2PE | 2-Methyl pentane | 0.31±0.14 | NaN | 0.28±0.05 | 0.30±0.04 | 0.20±0.00 | 0.36±0.27 |
| *NC6H14 | n-Hexane | 0.15±0.11 | 0.08±0.01 | 0.15±0.04 | 0.10±0.03 | 0.05±0.00 | 0.28±0.28 |
| *NC7H16 | n-Heptane | 0.03±0.01 | NaN | 0.07±0.00 | NaN | NaN | 0.07±0.00 |
| *NC8H18 | n-Octane | 0.03±0.00 | NaN | 0.03±0.00 | NaN | NaN | 0.03±0.00 |
| *EBENZ | Ethyl Benzene | 0.05±0.04 | 0.02±0.01 | 0.05±0.01 | 0.05±0.02 | 0.01±0.00 | 0.08±0.09 |

| | | | | | | | |
|---|---|---|---|---|---|---|---|
| *MXYL | m-Xylene | 0.03±0.03 | 0.01±0.00 | 0.03±0.01 | 0.03±0.01 | 0.01±0.00 | 0.02±0.02 |
| *OXYL | o-Xylene | 0.04±0.03 | 0.01±0.00 | 0.03±0.01 | 0.03±0.01 | 0.01±0.00 | 0.03±0.03 |
| **CH2O2 | Formic acid | 1.02±0.44 | 0.58±0.08 | 1.03±0.19 | 1.16±0.20 | 1.55±0.11 | 1.54±0.47 |
| **C2H4O2 | Acetic acid | 2.76±1.46 | 1.59±0.34 | 3.03±0.68 | 4.54±0.35 | 3.19±0.61 | 4.38±3.25 |
| **C2H8O2 | Ethylene dihydrate | 0.06±0.02 | 0.06±0.00 | 0.06±0.01 | 0.05±0.00 | 0.04±0.00 | 0.09±0.06 |
| **C5H8 | Isoprene | 0.31±0.24 | 0.16±0.06 | 0.36±0.14 | 0.69±0.46 | 0.56±0.33 | 0.54±0.25 |
| **C4H6O | Methyl Vinyl Ketone+ Methacrolein | 0.16±0.10 | 0.06±0.01 | 0.22±0.06 | 0.26±0.05 | 0.15±0.06 | 0.32±0.19 |
| **C3H4O2 | Acrylic acid | 0.12±0.05 | 0.06±0.01 | 0.13±0.03 | 0.16±0.02 | 0.13±0.02 | 0.19±0.10 |
| **C3H6O2 | Propanoic acid/ Hydroxy acetone | 0.90±0.43 | 0.57±0.15 | 0.97±0.23 | 1.26±0.03 | 1.01±0.11 | 1.45±0.93 |
| **C6H6 | Benzene | 0.28±0.13 | 0.12±0.03 | 0.33±0.03 | 0.43±0.04 | 0.25±0.05 | 0.38±0.21 |
| **C6H12 | Cyclohexane | 0.02±0.01 | 0.01±0.00 | 0.03±0.00 | 0.03±0.01 | 0.02±0.01 | 0.04±0.03 |
| **C3H4O3 | Pyruvic acid | 0.05±0.02 | 0.03±0.00 | 0.05±0.01 | 0.07±0.01 | 0.07±0.00 | 0.06±0.03 |
| **C7H8 | Toluene | 0.38±0.27 | 0.20±0.10 | 0.46±0.11 | 0.50±0.08 | 0.24±0.04 | 0.69±0.67 |
| **C8H10 | Xylene | 0.25±0.22 | 0.09±0.08 | 0.35±0.07 | 0.49±0.17 | 0.07±0.05 | 0.41±0.34 |
| **C10H16 | Monoterpene | 0.05±0.03 | 0.03±0.00 | 0.06±0.01 | 0.10±0.06 | 0.09±0.04 | 0.07±0.03 |
| **CH2O | Formaldehyde | 1.03±0.41 | 0.62±0.05 | 1.17±0.11 | 1.72±0.10 | 1.59±0.17 | 1.17±0.42 |
| **C2H4O | Acetaldehyde | 1.88±0.90 | 0.98±0.13 | 2.10±0.41 | 2.74±0.16 | 1.96±0.36 | 3.17±1.98 |
| **C3H6O | Acetone | 3.88±1.60 | 2.18±0.31 | 4.43±0.74 | 5.64±0.49 | 5.91±0.47 | 5.92±2.85 |
| **C3H4O | Acrolein | 0.25±0.11 | 0.14±0.02 | 0.29±0.05 | 0.39±0.04 | 0.33±0.05 | 0.39±0.19 |
| **C4H8O | MEK + Butanals | 0.45±0.30 | 0.24±0.04 | 0.53±0.16 | 0.59±0.05 | 0.44±0.05 | 0.87±0.86 |
| **C8H8O | Methyl benzaldehyde | 0.04±0.03 | 0.02±0.00 | 0.05±0.01 | 0.06±0.00 | 0.04±0.01 | 0.08±0.06 |
| &BVOC | Biogenic VOCs | 0.3±0.4 | 0.2±0.1 | 0.4±0.1 | 1.1±0.6 | 0.8±0.4 | 0.7±0.5 |
| &AVOC | Anthropogenic VOCs | 7.1±3.6 | 4.0±0.7 | 7.6±0.9 | 7.7±0.9 | 4.4±0.7 | 11.1±8.7 |
| &OVOC | Oxygenated VOCS | 7.2±7.4 | 7.0±1.0 | 9.2±1.5 | 18.6±1.3 | 16.4±1.9 | 14.9±12.8 |
| &Arom | Aromatic compounds | 0.6±0.6 | 0.4±0.2 | 0.8±0.1 | 1.5±0.3 | 0.6±0.1 | 1.2±1.3 |
| &Alkane | Alkane | 6.5±3.4 | 3.6±0.5 | 6.8±0.8 | 6.3±0.6 | 3.8±0.6 | 9.9±7.5 |
| &Alkene | Alkene | 2.5±1.9 | 0.5±0.1 | 2.2±0.2 | 2.6±0.2 | 1.0±0.5 | 2.5±1.0 |
| &Aldehyde | Aldehyde | 4.4±4.5 | 4.2±0.5 | 5.7±0.8 | 11.4±0.8 | 10.4±1.1 | 9.1±7.6 |
| &Acid | Acid | 2.8±2.9 | 2.8±0.5 | 3.4±0.6 | 7.2±0.5 | 5.9±0.8 | 5.8±5.2 |

Notes:
The concentration unit is presented in the bracket in the 'Species Abb.'; the unit for other species is ppb.
* Species measured by GC-MS and constrained by the model.
** Species measured by PTR-MS and constrained by mode.
[†] Species measured by instrument specified in Table 1.
[&] Different VOC functional groups.

[revised manuscript text omitted]

---

## Author Comment (AC2)

Thank you for your helpful comments our manuscript (EGUSPHERE-2022-854). The comments lead us to re-examine the simulation and observation results and the discussions we had made. Please find our itemized response below and the corrections in the re-submitted files. The original comments are in **black.** The responses to the comments are in **blue**. The changes in the manuscript and the supplementary were highlighted in **yellow** and were cited in **purple** in this response.

Comments #2

This paper presented field measurements conducted and a coastal site in Hong Kong from October to November 2020, including OH radical and multiple trace gases. OH, measurement was deployed by a new instrument using Chemical Ionization Mass Spectrometry (CIMS). The maximum OH concentration was $1.5 \times 10^7$ cm^-3 on 7 November, which are on the higher end of the measurements obtained in the PRD region. Two groups of air parcels were identified during the campaign, i.e. continental air and coastal air. The OH concentrations were generally higher in continental air (maximum of diurnal average: $5 \times 10^6$ cm^-3) compared to that of the coastal air (maximum of diurnal average: $4 \times 10^6$ cm^-3), so did for most of the trace gases, e.g. NO, NO2, HONO, VOCs (except O3). A box model based was used to simulate OH concentration, which could reproduce the OH concentration for continental air but overpredicts for coastal air. The overprediction was attributed to missing OH reactivity. As explained by the authors, OH measurement is difficult and additional measurement certainly helps to enrich the data set. Also, the authors made large efforts to review the published results presenting a nice overview of the current OH measurement. The structure may be improved if the authors could balance the introduction and results/discussion. Nevertheless, this reviewer suggests the publication of this paper once the following comments are addressed.

Response: Thanks for your comments and encouragement. The revised manuscript contained a more comprehensive discussion based on the comments of the reviewers which balance the structure of the article.

**General comments:**

As the instrument was deployed/presented for the first time, the description of the OH instrument is not sufficient. Although the authors referred to the technical details in a preprint (Pu et al. 2020 AMTD), this preprint has not been finally published and readers may have difficulties understanding some of the unaddressed issues. For example, the HO2+NOàOH+NO2 reaction that occurs in the inlet could be a positive bias to the OH measurement, which cannot be subtracted by the scavenge procedure. In the preprint, it's said that the reaction time was reduced to 47ms to minimize interference. However, it will produce a similar amount of OH in this reaction time at 1 ppb of NO and HO2 about 2 orders of magnitude higher than OH.

Response: Thanks for the comments on need to add more details on the CIMS. A similar comment was raised by referee 1. As indicated in our response to referee 1, our preprint in AMT did not move to the full publication stage as one referee of the AMT preprint thought our CIMS did not outperform the CIMS in other groups. Following the suggestions from both referees for the present work, we have moved some of the content in the AMTD preprint to the supplementary to support the measurements in this study. See response to referee 1 for details in added materials.

On the positive bias from the reaction of $HO_2$+ NO in the inlet, Tanner et al., (1997) calculated the OH produced by the $HO_2$ recycling reaction under different NO conditions (from <60 ppt to 1-2 ppb) in the inlet by a box model. Their results showed that the positive bias of less than $0.5 \times 10^6$ cm$^{-3}$ with a 60 ms conversion time, and the bias does not increase with the increase of NO concentration. In our study, a shorter (47ms) conversion time was used, we thus think positive bias is not large (<10%), but should be quantified in the future. However, such positive bias would increase the model-measurement discrepancy in the coastal air mass.

The discussion for the positive bias in the inlet by NO+$HO_2$ was added in supplementary as shown below:

"The reaction time affects the positive bias of OH arising from $HO_2$ + NO in the inlet. To estimate this bias, Tanner et al., (1997) calculated the OH produced by the $HO_2$ recycling reaction under different NO conditions (from < 60 ppt to 1-2 ppb) in the inlet by a box model. Their results showed that the positive bias of less than $0.5 \times 10^6$ cm$^{-3}$ with a 60 ms conversion time, and the bias does not increase with the increase of NO concentration. Thus, the conversion time of 47 ms in our study should further reduce such positive bias."

The box model calculation overestimated the observed OH concentration by 73% for coastal air. The authors attributed the model-measurement discrepancy to missing reactivity. However, more discussion/explanation is needed to explore all possibilities. The justification for missing reactivity is not sufficient. In Line 11 Page 9, it's described that the VOC below the detection limit was set to half of the detection limit, which seems arbitrary. How sensitive are the model results depending on this assumption? The derived missing reactivity only appeared in the coastal air masses but not the relatively more polluted air masses from the continental sector, which is a contradicting feature and needs more explanation.

Response: Thanks for the comments. More discussion and explanation on measurement and simulation uncertainties and $HO_2$ budgets were added to the revised manuscript (see below). After considering all possibilities, we still incline to attribute the discrepancy to missing reactivity.

"**3.3.1 Uncertainties in OH measurement and simulation**
The OH measurement uncertainties have been calculated as described in Section 2.2

and are shown as the error bars in Figures 5 and 6. The model's overestimation of OH in coastal air masses exceeded the measurement uncertainties (Figures 6 and 7), and thus, the measurement uncertainty is unlikely to be the main reason for the discrepancy.

Model uncertainties in our study include the uncertainties in photolysis frequencies correction, uncertainties in the constrained VOCs concentrations when they were below detection limits, and uncertainties from not considering halogen chemistry. On the first possibility, we acknowledge that the correction factor for photolysis frequencies due to cloud presence may be different for different species (Walker et al., 2022), thus, using a single correction factor (based on $J_{NO2}$) may introduce uncertainty in the model simulations. We think such uncertainty should not be significant because the weather was mostly sunny in the coastal cases. Regarding the uncertainty from the VOCs input, we conducted a sensitivity test to show that the treatment of VOCs that were below the detection limits should have a negligible effect on OH simulation ($RUN_{VOC0}$ and $RUN_{VOCDL}$ in Figure S4). We did not include halogen chemistry in our study as we wanted to compare our results with previous modelling work most of which did not consider halogen chemistry. Our other studies at the same site that did consider the halogen chemistry show a 4% increase in OH concentration from Cl chemistry (Peng et al., 2022) and 2.8% from Br chemistry (Xia et al., 2022), which would even increase the model-measurement discrepancy.

**3.3.2 Overestimation of OH sources**
Our calculated OH budgets show that the main sources of OH in the coastal air masses were the $HO_2 + NO$ reaction (69%), $O_3$ photolysis (14%), HONO photolysis (7%), and the reaction between ozone and $HO_2$ (4%). In the simulation, NO, HONO, and $O_3$ were constrained by observations. Could $HO_2$ be overestimated which would cause overprediction of OH?

The main $HO_2$ sources are the VOCs oxidation by OH and the photolysis of OVOCs. In our study, VOCs and OVOCs were more likely under-measured than over-measured, which would underpredict $HO_2$ rather than overpredicting it. In addition, not including the halogen chemistry would under-simulate $HO_2$ at this site (Peng et al., 2022; Xia et al., 2022).

We next examine the possibility of the underestimation of $HO_2$ sinks as the cause of the overprediction of OH. The major sinks of $HO_2$ include the reaction of NO to recycle OH, self-reaction to form $H_2O_2$, and heterogeneous loss by aerosol uptake. The first and second pathways have been considered in the MCM. The heterogeneous uptake … …"

The sensitivity test for the VOCs below the detection limit was added in the supplementary Figure S4 below. In short, those VOCs below detection limits have a negligible effect on the discrepancy. Even if we use the detection limits as model input, the simulation of OH concentration has little changes.

[Figure]

"Figure S4. Sensitivity tests for the simulated OH and HO$_2$ in continental and coastal cases and on 10 October. RUN$\gamma_{MAX}$ shows the simulated results for the maximum heterogeneous uptake effect of HO$_2$ ($\gamma = 1$). The RUN$_{VOC0}$ and RUN$_{VOCDL}$ show the simulated results that constraints "0" and the detection limit value as the concentration of VOCs when their concentration were below detection limits."

We agree that the discussion should focus more on the contradicting feature that the missing reactivity appeared in coastal air masses. We believe that this is an indicator of our knowledge gaps in coastal air mass. The K$_{miss}$ discussion in previous manuscript aims to figure out why discrepancy presents in the coastal case only. This discussion remains in the revised updated manuscript. The changes were shown below:

"We next explored the dependence of $k_{miss}$ on different trace gases. Figure 9a shows the correlation between $k_{miss}$ and NO concentration for the nine case days (including 10 October) between 09:00 and 15:00. At NO > 0.5 ppb, k$_{miss}$ is close to zero. At NO < 0.5 ppb, k$_{miss}$ tended to increase with decreasing NO. Similarly, k$_{miss}$ approached zero at high concentrations of NO$_2$ (> 2.5 ppb), TEXs (> 0.25 ppb), and AVOCs (> 5 ppb; Figure 9) and increased with decreasing concentrations of NO$_2$, TEXs, and AVOCs. High k$_{miss}$ also typically occurred at low toluene/benzene ratios and low C$_2$H$_2$/CO ratios (Figure 9), which are indicators of an aged air mass (Xiao et al., 2007; Kuyper et al., 2020).

Therefore, while we cannot completely rule out other possibilities, we argue that the aged coastal air masses could have contained unmeasured species such as oxygenated organic molecules (OOMs; Nie et al., 2022) and ocean-emitted gases (Thames et al., 2020) that contributed to the missing OH reactivity, causing the model to overestimate OH concentrations on the coastal case days. "

How to prove missing OH reactivity for the coastal air masses. On the other hand, discussion on the other possibility of OH model-measurement discrepancy is missing. For example, the OH measurement interference, if existed, could be proportional to the ambient NO level, which could lead to a more significant biased in the continental air. In this case, the model could consistently overpredict OH concentration for both air masses.

Response: Thanks for the comments. As discussed in our response to the general comment, we don't think the positive bias in the inlet due to reaction NO +HO2 would be significant.

As the campaign was conducted at a coastal site, the role of halogen chemistry was not mentioned. It's not clear if the FOAM model contains halogen chemistry. But the original MCM only contains simple Cl reactions with alkanes. It's suggested to include other key halogen chemistry in the box model. Or if the halogen chemistry is not important, more explanation is needed.

Response: We did not include halogen chemistry in our model as we wanted to compare our results with previous modelling work, most of which did not consider halogen chemistry. Our other studies at the same site that did consider the halogen chemistry show a 4% increase in OH concentration from Cl chemistry (Peng et al., 2022) and 2.8% from Br chemistry (Xia et al., 2022), which would even increase the model-measurement discrepancy in the coastal air.

The discussion and citation were added in the updated version as shown below:

"We did not include halogen chemistry in our study as we wanted to compare our results with previous modelling work most of which did not consider halogen chemistry. Our other studies at the same site that did consider the halogen chemistry show a 4% increase in OH concentration from Cl chemistry (Peng et al., 2022) and 2.8% from Br chemistry (Xia et al., 2022), which would even increase the model-measurement discrepancy in the coastal air mass."

Technical comments:

Introduction: It's not clear why isoprene chemistry was discussed here.

Response: The isoprene chemistry partially explains the missing sources of OH in the low NO-high VOCs forest areas in previous observation done by LIF, this we think it is relevant to have a discussion to complete the "map of OH discrepancy".

Line 4-6 Page 10: It's strange to denote the maximum of different compounds while some of these are clearly anticorrelated (e.g. O3 and NO2).

Response: the purpose of the denotation of the maximum of different compounds is to provide the exact maximum value to the reader instead of describing the anticorrelation.

The revised contents are shown as below:

"As a primary source of OH, HONO peaked in the morning at $0.21 \pm 0.09$ ppb around 7:00 local time (LT), and $O_3$ peaked in the afternoon at $70 \pm 20$ ppb at around 16:00 LT."

Line 11 Page 10: $(4.0 \pm 2.1)$

Response: We believed this should be "$4.9 \pm 2.1$" instead of "4.0".

Line 23 Page 13: "fake" suggested to be artificial loss

Response: Thanks for the correction on terminology. We changed "a fake" to "an artificial loss"

**Reference:**

Kuyper, B., Wingrove, H., Lesch, T., Labuschagne, C., Say, D., Martin, D., Young, D., Khan, M. A. H., O'Doherty, S., Davies-Coleman, M. T., and Shallcross, D. E.: Atmospheric Toluene and Benzene Mole Fractions at Cape Town and Cape Point and an Estimation of the Hydroxyl Radical Concentrations in the Air above the Cape Peninsula, South Africa, ACS Earth Space Chem., 4, 24–34, https://doi.org/10.1021/acsearthspacechem.9b00207, 2020.

Nie, W., Yan, C., Huang, D. D., Wang, Z., Liu, Y., Qiao, X., Guo, Y., Tian, L., Zheng, P., Xu, Z., Li, Y., Xu, Z., Qi, X., Sun, P., Wang, J., Zheng, F., Li, X., Yin, R., Dallenbach, K. R., Bianchi, F., Petäjä, T., Zhang, Y., Wang, M., Schervish, M., Wang, S., Qiao, L., Wang, Q., Zhou, M., Wang, H., Yu, C., Yao, D., Guo, H., Ye, P., Lee, S., Li, Y. J., Liu, Y., Chi, X., Kerminen, V.-M., Ehn, M., Donahue, N. M., Wang, T., Huang, C., Kulmala, M., Worsnop, D., Jiang, J., and Ding, A.: Secondary organic aerosol formed by condensing anthropogenic vapours over China's megacities, Nat. Geosci., 15, 255–261, https://doi.org/10.1038/s41561-022-00922-5, 2022.

Peng, X., Wang, T., Wang, W., Ravishankara, A. R., George, C., Xia, M., Cai, M., Li, Q., Salvador, C. M., Lau, C., Lyu, X., Poon, C. N., Mellouki, A., Mu, Y., Hallquist, M., Saiz-Lopez, A., Guo, H., Herrmann, H., Yu, C., Dai, J., Wang, Y., Wang, X., Yu, A., Leung, K., Lee, S., and Chen, J.: Photodissociation of particulate nitrate as a source of daytime tropospheric $Cl_2$, Nat Commun, 13, 939, https://doi.org/10.1038/s41467-022-28383-9, 2022.

Tanner, D. J., Jefferson, A., and Eisele, F. L.: Selected ion chemical ionization mass

spectrometric measurement of OH, J. Geophys. Res., 102, 6415–6425, https://doi.org/10.1029/96JD03919, 1997.

Thames, A. B., Brune, W. H., Miller, D. O., Allen, H. M., Apel, E. C., Blake, D. R., Bui, T. P., Commane, R., Crounse, J. D., Daube, B. C., Diskin, G. S., DiGangi, J. P., Elkins, J. W., Hall, S. R., Hanisco, T. F., Hannun, R. A., Hintsa, E., Hornbrook, R. S., Kim, M. J., McKain, K., Moore, F. L., Nicely, J. M., Peischl, J., Ryerson, T. B., St. Clair, J. M., Sweeney, C., Teng, A., Thompson, C. R., Ullmann, K., Wennberg, P. O., and Wolfe, G. M.: Missing OH reactivity in the global marine boundary layer, Atmospheric Chemistry and Physics, 20, 4013–4029, https://doi.org/10.5194/acp-20-4013-2020, 2020.

Walker, H. L., Heal, M. R., Braban, C. F., Whalley, L. K., and Twigg, M. M.: Evaluation of local measurement-driven adjustments of modelled cloud-free atmospheric photolysis rate coefficients, Environ. Sci.: Atmos., 2, 1411–1427, https://doi.org/10.1039/D2EA00072E, 2022.

Xia, M., Wang, T., Wang, Z., Chen, Y., Peng, X., Huo, Y., Wang, W., Yuan, Q., Jiang, Y., Guo, H., Lau, C., Leung, K., Yu, A., and Lee, S.: Pollution-Derived $Br_2$ Boosts Oxidation Power of the Coastal Atmosphere, Environ. Sci. Technol., 56, 12055–12065, https://doi.org/10.1021/acs.est.2c02434, 2022.

Xiao, Y., Jacob, D. J., and Turquety, S.: Atmospheric acetylene and its relationship with CO as an indicator of air mass age, J. Geophys. Res., 112, D12305, https://doi.org/10.1029/2006JD008268, 2007.

---

## Author Response (AR2)

Thank you for your second-round comments on our manuscript (EGUSPHERE-2022-854). Please find our itemized responses below and the corrections in the re-submitted files. The original comments are in **black.** The responses to the comments are in **blue**. The changes in the manuscript and the supplementary were highlighted in **yellow** and were cited in **purple** in this response.

The revised paper is very much improved, and the authors have responded satisfactorily to my questions and comments. The line of argumentation why unmeasured OH reactants are most likely responsible for the model overestimation of OH is now plausible. I recommend publication of the manuscript after minor revisions.

Response: Thanks for the encouraging comments.

The authors should extend the discussion on missing OH reactivity, as they consider missing OH sinks important enough to mention in the title. The authors estimate the missing reactivity that would be needed to match the modeled OH to the observation. The estimate assumes that the unknown reactant(s) consume OH without subsequent OH recycling. In the lower troposphere, the only relevant OH sink that acts as a radical termination reaction is the reaction of OH with NO2, which is already included in the model. All other reactions produce either HO2 or RO2 which can recycle OH when there is NO present. I suggest that the authors perform sensitivity tests where they include OH recycling when they adjust the missing OH reactivity. They could assume a recycling mechanism for the unknown reactant that behaves, for example, like for CH4 or CH3CHO, where the formed RO2 is partially recycled back to OH with NO. How much additional OH reactivity would be needed in these cases? Can the authors speculate in more detail about the possible origin of the missing reactant(s)?

Response: Thanks for the suggestion on the sensitivity test. We think the suspected missing VOCs should be quite reactive, thus we chose $CH_3CHO$ suggested by the referee for the sensitivity test. Results show that, if the recycling mechanism is considered, on average, additional 19.5 ppb $CH_3CHO$ would be needed as the additional OH sink to match the observation, with a calculated OH reactivity of 7.2 $s^{-1}$, compared to the calculated $k_{miss}$ in section 3.3.3 (5.0 $s^{-1}$) without recycling. We have added additional discussion in section 3.3.3 and a revised Figure S4 as shown below:

"We conducted a sensitivity test in which we assume the missing sink is resulting from under-measured $CH_3CHO$. Results show that $CH_3CHO$ concentrations would increase by 20 times ($RUN_{CH3CHO}$) to make up the missing OH sinks and the missing reactivity with cycling of $CH_3CHO$ oxidation product would increase to 7.2 $s^{-1}$ (Figure S4)."

"

[Figure]

Figure S4. Sensitivity tests for the simulated OH and HO$_2$ in continental and coastal cases on 10 October. RUN$_{CH3CHO}$ shows the simulated results of the selected coastal cases when additional CH$_3$CHO is added as OH sinks. RUN$\gamma_{MAX}$ shows the simulated results for the maximum heterogeneous uptake effect of HO$_2$ ($\gamma = 1$). The RUN$_{VOC0}$ and RUN$_{VOCDL}$ show the simulated results that constraints "0" and the detection limit value as the concentration of VOCs when their concentration was below detection limits."

additional OH reactivity would be needed
Specific comments.

1. Page 1 - Line 25. Missing OH reactivity is generally defined as a pseudo-first order rate coefficient for OH loss by unmeasured reactants.

Response: Thanks for the comments, we agree to the definition and revised the sentence as follows:

"A missing OH reactivity, which is defined as the pseudo-first-order rate coefficient for OH loss by unmeasured trace gases was estimated as $5.0 \pm 2.6$ s$^{-1}$……"

2. Page 1 - Line 26 – 28. The sentence is incomprehensible.

Response: Thanks for the comments, we revised the sentence as follows:

"Unaccounted-for OH sinks in the model are proposed to be the cause of this overprediction. A missing OH reactivity, which is defined as the pseudo-first-order rate coefficient for OH loss by unmeasured trace gases was estimated as $5.0 \pm 2.6$ s$^{-1}$ (lower limit) in the coastal air, and the missing reactivity increased with decreasing concentrations of NO$_x$ and volatile organic

compounds (VOCs)."

3. Page 2 - Line 18 – 19. "… tropospheric OH radicals can now be detected…". Sounds strange. These techniques have been measuring atmospheric OH for 30 years.

Response: We deleted "now" in the revised sentence as follows:

"Through decades of efforts, tropospheric OH radicals have been successfully detected following the development of ……"

4. Page 3 – Line 27 – 28. References missing.

Response: The review article (Heard and Pilling, 2003) was added as a reference.

5. Page 4 – Line 2 – 4. The development of the LIM0 mechanism was initially inspired by unexplained high OH concentrations observed during field campaigns in forested regions. Very soon after, laboratory and chamber experiments paved the way for further improvements in isoprene mechanisms that are further developments of LIM0 (for an overview, see Novelli et al., 2020 and Wennberg et al., Chem. Rev. 2018, 118, 3337−3390).

Response: Thanks for the comments. Since the missing sources and the LIM mechanism are not the focus of our article, we reduced the discussion on details of the mechanism and rewrote the relevant content as below:

"The Leuven isoprene mechanism (LIM) was then developed to explain the high OH concentrations observed during field campaigns in forested regions, based on laboratory and chamber experiments of isoprene oxidation (Wennberg et al., 2018; Novelli et al., 2020). With the adoption of this mechanism, ……"

6. Page 7 – Line 6. The sentence should end with ", respectively".

Response: Thanks for the comments, the sentence ended with ", respectively" now.

7. Page 8 – Line 10. "… at 62…". Typo?

Response: Thanks for the careful check, "at 62" is the correct description. However, the label for it should be $S_{62}$ instead of $S_{64}$. Based on this comment, the rest of "$S_{64}$" was changed to $S_{62}$ in the manuscript and supplementary for consistency.

8. Page 8 – Line 23. "… Table S1…". Table 1?

Response: Thanks for the careful check, it should be Table 1.

9. Page 9, Equation E4. A lower-case symbol (e.g., v) should be used for the mean molecular

velocity. The symbol for molar mass is generally M. Thus, M_HO2 would be suitable.

Response: Thanks for the suggestion. The equation has been changed as follows:

$$"v_{HO_2} = \sqrt{\frac{8RT}{\pi \times M_{HO_2}}} \ (E4)"$$

10. Page 10 – Line 12 – 13. It is surprising that the correlation with jO1D is so much worse than with jNO2. How do these results compare with other observations in coastal regions (for example, Berresheim et al., Atmos. Chem. Phys., 3, 639–649, 2003)?

Response: We appreciate the reviewer's comment. First, the relationships for OH versus $J_{O1D}$ and OH versus $J_{NO2}$ are not always linear under different $NO_x$ concentrations (Berresheim et al., 2003; Rohrer and Berresheim, 2006). These relationships become more complicated when $O_3$ photolysis is not the major $HO_x$ source (Berresheim et al., 2003), as is the case in our study (Table 2). Therefore, we do not compare the linear fit $R^2$ values for OH versus $J_{O1D}$ and OH versus $J_{NO2}$.

In addition, the relationship between OH and $J_{O1D}$ has been used to learn about OH production mechanisms when $O_3$ photolysis is the dominant $HO_x$ source (Berresheim et al., 2003). This is not neccesarily applicable in our study when HONO photolysis is 50-70% as strong as $O_3$ photolysis as a $HO_x$ source (Table 2). Instead, we use the F0AM model with more complete mechanisms to learn about OH production. We show the figures for OH versus $J_{O1D}$ and OH versus $J_{NO2}$ (Figure. S2) just to demonstate that our OH measurements are reasonable. The changes were shown below:

"The OH concentrations showed a distinct diurnal pattern and a positive correlation with $J_{NO2}$ ($R^2 = 0.68$) and calculated $J_{O1D}$ ($R^2=0.46$) (Figure S2), similar to the findings in previous studies (Berresheim et al., 2003; Rohrer and Berresheim, 2006; Ma et al., 2019)."

11. Page 10 – Lines 17 – 18. "total measured VOCs reactivity" may be misunderstood as measured reactivity of VOCs. It would be better to say: total reactivity of measured VOCs.

Response: Thanks for the comment, the description for reactivity of VOCs and OVOCs was changed as recommended.

12. Page 11 – Lines 5 – 10. I am surprised about the levels of NO (~ 0.1 ppb) and isoprene (~0.1 ppbv) at night shown in Fig. 4. At the given O3 mixing ratio of about 40 ppbv and 0.5 x 10^6 OH/cm3, the expected lifetime of NO and isoprene is 1 min and 4 h, respectively. Is anything known about the sources of NO and isoprene that maintain the observed concentration levels at night?

Response: The nighttime isoprene might be due to the nocturnal emission from the plants surrounding the site. Since the isoprene emission from the plants is closely related to the temperature (Miyama et al., 2013), and most of the nighttime temperature in this campaign was above 20°C (Figure 3), we believe that the nocturnal emission of isoprene from the plant are nonnegligible.

The nighttime NO was due to the emissions from the ships as reported in previous studies (Wang et al., 2019).

The discussion were added as follow:

"Non-negligible levels of NO (~ 0.1 ppb) and isoprene (~ 0.1 ppb) were observed at night, which could be caused by nearby ship emissions and plant emissions, respectively."

13. Page 13 – Line 2. The symbol p should be defined.

Response: Thanks for the comments, the changes were shown below:

"The coastal air masses showed statistically significant (i.e., $p$-value < 0.05) lower $NO_x$ (−63%), AVOCs (−47%), BVOCs (−50%), OVOCs (−23%), and CO (−31%) concentrations compared with the continental cases (Figure 7, Table S3). (The $p$-value is the probability of the difference of two data sets occurs by chance)."

P14. Page 16 – Line 15. "… possible missing source…". Do you mean missing sink?

Response: Thanks for the careful check, it should be missing sinks.

15. Page 30, Figure 3. What is the meaning of M in "VOCsReac(M)"? The expression looks like a function on M.

Response: "(M)" denotes "measured". To avoid confusion, "(M)" was removed from Figures 3,4,7 and 8.

16. Page 30, Figure 3. A horizontal dotted line at OH = 0 would be helpful to understand the diurnal profile of OH.

Response: Thanks for the suggestion, A horizontal dotted line was added, and the revised figure is shown below:

[Figure]

"Figure 3. Time series of OH between 7 October and 23 November with measured weather conditions (temperature and RH), OH primary sources (ozone and HONO), $NO_x$ (NO and $NO_2$), the reactivity of measured VOCs and OVOCs ($VOCs_{Reac}$ and $OVOCs_{Reac}$), and photolysis frequency ($J_{NO2}$). All measurement data shown are 10 min averages. The gaps in the data were due to calibration or instrument maintenance. The black lines separate the non-continuous days during measurement and the black horizontal dotted line denotes [OH]=0. The grey shaded area denotes night-time. The time zone was the local time (+8 UTC) for the x-axis."

Supplement in egusphere-2022-854-ATC1.pdf

1. Page 41 – Line 2. Figure 3 must be Figure 2 ?

Response: Thanks for the careful check, it should be Figure 2.

2. Page 41 – Line 12. The quantity "It" is a photon area density (photons/cm2), sometimes called "photon fluence". Please check unit in Table S2.

Response: Thanks for the comment. The unit of "It" in Table S2 was changed to photons/$cm^2$. Please see the revised Table S2 in the response for item 10.

3. Page 41 – Eq. SE2. What are the units of TS97, BS96 and S64? In Table S2, the calibration factor C is given in units of OH·cm3/Hz. Please explain!

Response: Thanks for the careful check, the unit of TS97, BS97, and S64 is Hz. However, the Hz were cancelled out in the calculation of the calibration factor (SE2: $C = \frac{1}{[OH]_{CAL}} \times \frac{TS_{97} - BS_{97}}{S_{62}}$ ). Therefore, the unit of calibration factor (C) should be $cm^3$.

Also, as mentioned above, the "$S_{64}$" should be "$S_{62}$" instead. Please see the revised Table S2 in

the response for item 10.

4. Page 41 – Eq. SE3. What about statistical noise of S64? Shouldn't it be included in the calculation of DL?

Response: Thanks for the comment. We should consider the noise of $S_{62}$ in DL claculation. The corrected equation is shown below:

"$DL = \frac{1}{C} \times n * \sigma(\frac{BS_{97}}{S_{62}})$

Where DL is the detection limit in cm$^{-3}$, $C$ is the calibration factor, and $n$ is the ratio of signal to noise S/N. $\sigma(\frac{BS_{97}}{S_{62}})$ represents the standard deviation of the background measurement when the scavenger was added through front injectors. The detection limit (n = 3, average time = 6 minutes) in the laboratory was approximately $1.7 \times 10^5$ molecules cm$^{-3}$ on average."

Based on this equation, the calculated DL will changed from $1.5 \times 10^5$ to $1.7 \times 10^5$ in lab conditions. The DL for daytime and nighttime will changed from $10 \times 10^5$ to $12 \times 10^5$ and $7.7 \times 10^5$ to $8.5 \times 10^5$, respectively. Please see the change of detection limits in the revised Table S2 in the response for item 10.

5. Page 41. How large are typical signals (TS97, BS96 and S64)?

The $TS_{97}$ and $BS_{97}$ varied from tens to hundreds depending on the ambient chemical conditions, the strength of the reagent ion ($S_{62}$) signal, and the instrument status. The $S_{62}$ varied from tens of thousands to 200,000. The $S_{62}$ could change due to the inlets assembling and change in flows of injection gases and voltages of the CIMS, and the detector aging. That is why we use the ratio between them instead of their absolute signals of $TS_{97}$ and $BS_{97}$ for concentration calculation.

6. Page 42 – Line 1. I can't find the information in Figure 8. Wrong/missing Figure?

Response: Thanks for the careful check, Figure 8 is the wrong figure that should not be included in the manuscript. Please see the changes below:

"The detection limit (S/N = 3, average time = 6 minutes) in the laboratory was approximately $1.7 \times 10^5$ molecule cm$^{-3}$ on average. "

7. Page 42 – Line 25. What is the difference between S (sensitivity) and C (Eq. SE2)?

In our study, they both refer to the numerical presentation of the instrument's sensitivity. To avoid confusion, we now use C instead of S in the sensitivity optimization section.

8. Page 43 – Line 23. "C3F8" Typo?

Response: Thanks for the careful check, it should be $C_3H_8$.

9. Page 43 – Line 23 - 25. I do not understand what the authors mean. Please give a definition of "elimination rate". Why is it larger for C3F6 than for C3H8? Why does the purity of the gas play a role? What is meant by "stable quality"?

Response: Thanks for the comments. We changed the "elimination rate" to "scavenging efficiency (SE)" which is defined as $SE = \frac{TS97 - BS97}{TS97} \times 100\%$. We tested three cylinders of $C_3H_8$ during optimization, and they gave widely different SE although they are labelled in the same $C_3H_8$ concentration. We think that the purity of those $C_3H_8$ cylinder gases was questionable. Stable quality (for $C_3F_6$) means that compare to $C_3H_8$, $C_3F_6$ has a relatively consistent SE (>90%) for all cylinder gases we used regardless of supplier. Please see the changes below:

"We tested both $C_3F_6$ and $C_3H_8$ as scavenger gas for OH. We found that the $C_3H_8$ provided by our suppliers were questionable because the scavenging efficiency ($SE = \frac{TS97 - BS97}{TS97} \times 100\%$) by $C_3H_8$ in the different cylinders and different suppliers varied from 30% to 98% although the cylinders were labelled with the same concentration. In contrast, $C_3F_6$ from different cylinders labelled with the same concentration gave consistent SE. Therefore, we chose $C_3F_6$ as the OH scavenger gas."

10. Page 45, Table S2. Check the given detection limits (missing units!).

Response: Thanks for the careful check, the unit ($\times 10^5$ cm$^{-3}$) was added to the detection limits in Table S2. The revised table S2 is shown below:

| Efficiency Related | Parameters | Gas | Values | Units | Specification for Measurement | Values | Units |
|---|---|---|---|---|---|---|---|
| $E_{Conv}$ | Front Injection | $SO_2$ (0.9%) | 5 | sccm | Sample Flow [$SO_2$] | 12 | ppm |
| | Pulse Valve | $N_2$ | 2 | sccm | Cycle Duration (OH) | 6 | mins |
| | | $C_3F_6$ (99.9%) | 2 | sccm | Scavenging Efficiency (OH) | 92% | |
| | Rear Injection | $C_3F_6$ (99.9%) | 2 | sccm | Sample Flow [$C_3F_6$] | 1072 | ppm |
| | | $HNO_3$ | 10 | sccm | Reaction Time | 47 | ms |
| | Sample Flow | | 3.7 | slpm | Sample Flow Speed | 55 | cm/s |
| $E_{Ion}$ | Sheath Flow | Zero Air | 12.6 | slpm | Reynolds Number in Ionization Chamber | >4000 Turbulent flows | |
| | | $HNO_3$ | 10 | sccm | | | |
| | | $C_3F_6$ (99.9%) | 2 | sccm | Sheath Flow [$C_3F_6$] | 159 | ppm |
| | Total Flow | | 16.8 | slpm | Sheath Flow Speed | 25 | cm/s |
| | Sheath Voltages | | -80 | V | Voltages Difference for ionization | 48 | V |
| | Sample Voltages | | -32 | V | | | |
| $E_{Trans}$ | Buffer Gas | $N_2$ | 440 | sccm | Voltages Difference for transmission | 80 | V |
| | Buffer Voltages | | -70 | V | | | |
| | Pinhole Voltages | | -40 | V | | | |
| Cal | Calibration Flow | | 10 | slpm | Calibration Factor (Reagent ion: $NO_3^-$) | $1.21*10^{-8}$ | $cm^3$ |
| | Flow Speed | | 65 | cm/s | | | |
| | Product It Value | | $8.8*10^{10}$ | photon/cm$^2$ | | | |
| Uncertainties | Sigma (σ) | | 2 | | Detection Limit ($10^5 \times cm^{-3}$) | In the lab (3σ) | 1.7 |
| | Calibration | | 38% | | | Daytime (3σ) | 12 |
| | Overall | | 44% | | | Nighttime (3σ) | 8.5 |

11. Page 47, Table S3. For which time of day do the average values apply?

Response: The concentrations in Table S3 were averaged the daytime (6:00 to 18:00). The following sentence was added to the notes of the table:

"The concentration was averaged from the daytime (6:00 to 18:00) results."

Reference:

Berresheim, H., Plass-Dülmer, C., Elste, T., Mihalopoulos, N., and Rohrer, F.: OH in the coastal boundary layer of Crete during MINOS: Measurements and relationship with ozone photolysis,

Atmospheric Chemistry and Physics, 3, 639–649, https://doi.org/10.5194/acp-3-639-2003, 2003.

Heard, D. E. and Pilling, M. J.: Measurement of OH and HO$_2$ in the Troposphere, Chem. Rev., 103, 5163–5198, https://doi.org/10.1021/cr020522s, 2003.

Ma, X., Tan, Z., Lu, K., Yang, X., Liu, Y., Li, S., Li, X., Chen, S., Novelli, A., Cho, C., Zeng, L., Wahner, A., and Zhang, Y.: Winter photochemistry in Beijing: Observation and model simulation of OH and HO2 radicals at an urban site, Science of The Total Environment, 685, 85–95, https://doi.org/10.1016/j.scitotenv.2019.05.329, 2019.

Miyama, T., Okumura, M., Kominami, Y., Yoshimura, K., Ataka, M., and Tani, A.: Nocturnal isoprene emission from mature trees and diurnal acceleration of isoprene oxidation rates near Quercus serrata Thunb. leaves, Journal of Forest Research, 18, 4–12, https://doi.org/10.1007/s10310-012-0350-5, 2013.

Novelli, A., Vereecken, L., Bohn, B., Dorn, H.-P., Gkatzelis, G. I., Hofzumahaus, A., Holland, F., Reimer, D., Rohrer, F., Rosanka, S., Taraborrelli, D., Tillmann, R., Wegener, R., Yu, Z., Kiendler-Scharr, A., Wahner, A., and Fuchs, H.: Importance of isomerization reactions for OH radical regeneration from the photo-oxidation of isoprene investigated in the atmospheric simulation chamber SAPHIR, Atmos. Chem. Phys., 20, 3333–3355, https://doi.org/10.5194/acp-20-3333-2020, 2020.

Wang, T., Dai, J., Lam, K. S., Nan Poon, C., and Brasseur, G. P.: Twenty-Five Years of Lower Tropospheric Ozone Observations in Tropical East Asia: The Influence of Emissions and Weather Patterns, Geophysical Research Letters, 46, 11463–11470, https://doi.org/10.1029/2019GL084459, 2019.

Wennberg, P. O., Bates, K. H., Crounse, J. D., Dodson, L. G., McVay, R. C., Mertens, L. A., Nguyen, T. B., Praske, E., Schwantes, R. H., Smarte, M. D., St Clair, J. M., Teng, A. P., Zhang, X., and Seinfeld, J. H.: Gas-Phase Reactions of Isoprene and Its Major Oxidation Products, Chem. Rev., 118, 3337–3390, https://doi.org/10.1021/acs.chemrev.7b00439, 2018.